# Structural mechanism underlying G protein family-specific regulation of G protein-gated inwardly rectifying potassium channel

Hanaho Kano[1], Yuki Toyama [1], Shunsuke Imai[1], Yuta Iwahashi[1], Yoko Mase[1], Mariko Yokogawa[1,2], Masanori Osawa [1,2] & Ichio Shimada[1]

G protein-gated inwardly rectifying potassium channel (GIRK) plays a key role in regulating neurotransmission. GIRK is opened by the direct binding of the G protein βγ subunit (Gβγ), which is released from the heterotrimeric G protein (Gαβγ) upon the activation of G protein-coupled receptors (GPCRs). GIRK contributes to precise cellular responses by specifically and efficiently responding to the Gi/o-coupled GPCRs. However, the detailed mechanisms underlying this family-specific and efficient activation are largely unknown. Here, we investigate the structural mechanism underlying the Gi/o family-specific activation of GIRK, by combining cell-based BRET experiments and NMR analyses in a reconstituted membrane environment. We show that the interaction formed by the αA helix of Gαi/o mediates the formation of the Gαi/oβγ-GIRK complex, which is responsible for the family-specific activation of GIRK. We also present a model structure of the Gαi/oβγ-GIRK complex, which provides the molecular basis underlying the specific and efficient regulation of GIRK.

---

[1] Graduate School of Pharmaceutical Sciences, The University of Tokyo, Hongo, Bunkyo-ku, Tokyo 113-0033, Japan. [2] Present address: Faculty of Pharmacy, Keio University, Shibakoen, Minato-ku, Tokyo 105-8512, Japan. Correspondence and requests for materials should be addressed to I.S. (email: shimada@iw-nmr.f.u-tokyo.ac.jp)

Gprotein-gated inwardly rectifying potassium channel (GIRK) is a family of inwardly rectifying potassium channels that play important roles in regulating cellular excitabilities in the heart and brain[1]. GIRK opening is coupled to the activation of a G protein-coupled receptor (GPCR) on the cell surface. When a GPCR is stimulated by agonist binding, the activated GPCR catalyzes the nucleotide exchange reaction on heterotrimeric G protein (Gαβγ), in which the guanosine diphosphate (GDP) bound to the α subunit (Gα) is exchanged with guanosine triphosphate (GTP), and then Gαβγ dissociates into the active GTP-bound form of Gα (Gα(GTP)) and Gβγ[2]. GIRK is opened by directly binding to Gβγ released upon the activation of GPCRs, such as muscarinic acetylcholine receptors, γ-aminobutyric acid (GABA) receptors, dopamine receptors, and opioid receptors[3]. Under physiological conditions, the intracellular concentration of potassium ion (K$^+$) is maintained at a higher level than the extracellular K$^+$ concentration, and the resting membrane potential is held slightly above the equilibrium potential of K$^+$. Therefore, the outward K$^+$ current induced by the opening of GIRK hyperpolarizes the membrane and decreases cell excitabilities, thus regulating the heart rate and both the excitatory and inhibitory neurotransmissions. From a pharmacological viewpoint, GIRK is a potential therapeutic target for epilepsy and bipolar disorder[4,5].

One of the most important characteristics of GIRK is that its opening is dependent on the type of the extracellular stimulus. Under physiological conditions, GIRK activation is elicited by GPCRs that mediate inhibitory neurotransmission, such as GABA$_B$ receptors[6] and muscarinic M$_2$ receptor[7], but not by GPCRs in charge of stimulatory neurotransmission, such as β-adrenergic receptors[7]. This signal-specific response of GIRK prevents improper cross-talk between intracellular signaling pathways, and is thought to be essential for maintaining cellular homeostasis and neural activity[3,8]. The signal-specific response of GIRK is particularly puzzling, considering the fact that mammalian cells utilize the common G protein signaling machinery to respond to various extracellular stimuli[9], because Gβγ, the direct activator of GIRK, is also released in response to the activation of other GPCRs that are not related to GIRK activation. At the molecular level, the specificity of the GIRK activation has been explained by the finding that GIRK is exclusively opened by Gβγ, which is released upon the activation of GPCRs coupled to the Gαi/o family, and not by those released from GPCRs coupled to other Gα families (Gαq, Gαs, and Gα12/13)[10]. However, since Gαi/o is not a direct activator of GIRK and there are few functional differences among the Gβ$_{1-4}$γ subtypes[11,12], the detailed molecular mechanism for the Gαi/o-specific GIRK activation has long remained enigmatic.

The detailed mechanisms underlying the Gαi/o-specific and efficient activation of GIRK have been extensively characterized by electrophysiological analyses of GIRK[3,10,13,14], and fluorescence resonance energy transfer (FRET) and bioluminescence resonance energy transfer (BRET) analyses. From these studies, it has been proposed that the signaling complex, consisting of GPCR, GIRK, and Gαi/oβγ[15–17], is pre-formed in the cellular environment, and Gβγ released from Gαi/o rapidly and efficiently binds to GIRK within the same complex. However, since the resolution of the structural information is low due to the large size of the fluorescently labeled fusion proteins, questions still remain regarding which regions of the complex confer the Gαi/o family specificity and how the efficient and rapid activation of GIRK is accomplished while retaining the complex formation. For this point, there are several controversial models describing the signaling complex, in which the interactions formed between GIRK-Gα[18,19], GIRK-GPCR[17], or GIRK-Gβγ[20] enable the formation of the complex and determine the characteristics of GIRK activation.

In this research, we set out to characterize the direct interaction between Gαi/oβγ and GIRK, based on our previous finding that Gαi/o(GTP) interacts with the cytoplasmic region of GIRK[21]. Using cell-based BRET experiments, we demonstrate that the Gαi/oβγ–GIRK interaction is responsible for achieving the Gαi/o specificity in GIRK activation, and that the interaction between the helical domain of Gα and GIRK confers this specificity. To characterize the inherently weak Gαi/oβγ–GIRK interaction, we apply methyl-based Nuclear Magnetic Resonance (NMR) techniques[22], which show that Gαi/oβγ directly interacts with a chimeric GIRK channel consisting of the cytoplasmic region of mammalian GIRK1 and the transmembrane region of prokaryotic KirBac1.3. By utilizing NMR paramagnetic relaxation techniques, we successfully construct a model structure of the Gαi/oβγ–GIRK complex on the membrane, although the molecular weight of the complex, over 400 K, is far beyond the molecular weight limit of conventional NMR, and identify the key structural determinant of the selective binding between Gαi/oβγ and GIRK. From these results, we propose a mechanism for the Gαi/o-specific GIRK regulation that explains the rapid and efficient GIRK regulation in the physiological environment.

## Results

**The helical domain of Gα determines the family specificity.** First, we conducted cell-based assays to quantitatively evaluate the Gαi/o-specific activation of GIRK, to identify the structural element of Gα that determines the specificity in the activation of GIRK. To date, the Gαi/o specificity in regulating GIRK has been mainly characterized by electrophysiological analyses observing GPCR agonist-induced GIRK currents, using cultured cells or oocytes expressing a GPCR, GIRK, and various Gα mutants[10,13]. In these experiments, the Gαi/o specificity in activating GIRK was mainly observed as the differences in the steady-state of GIRK current upon the addition of GPCR agonists, indicating that the preference of Gαi/o is mainly under thermodynamic control, rather than the kinetic control. This steady-state GIRK-current has been compared between Gα families to characterize the Gαi/o specificity, however, the observed GIRK currents are strongly affected by the extents of G protein activation; i.e., the amounts of Gβγ released upon the activation of GPCRs, which can significantly differ among the Gα mutants analyzed. Therefore, the Gα specificity in activating GIRK over other effector proteins has been difficult to compare in a quantitative manner. Accordingly, we conducted cell-based BRET experiments, in which we monitored the thermodynamic stability of the binding of Gβγ to several effector proteins, including GIRK[23,24]. We conducted two sets of BRET experiments for each Gα protein: one to observe the intermolecular BRET between Gβγ and GIRK that reflects the GPCR-mediated GIRK activation, and the other to observe the intermolecular BRET between Gβγ and the Gβγ-binding domain of G protein-coupled receptor kinase (hereafter referred to as GRK), which does not exhibit a Gα family preference and serves as a reporter of the G protein activation[23–25]. By normalizing the BRET signals observed between Gβγ–GIRK with those observed between Gβγ–GRK, we can quantitatively compare the Gα specificity in the activation of GIRK between different Gα families and mutants.

Fig. 1a–d shows the schema and representative results of the BRET experiments, using NLuc-tagged GRK and GIRK (GRK-Luc and GIRK-Luc). The expression of G proteins and GIRK on the plasma membrane was confirmed by fluorescence imaging (Supplementary Fig. 1). In HEK293T cells expressing Venus-tagged Gβγ and GRK-Luc, along with delta opioid receptor (DOR) and Gα, the addition of Met-enkephalin (DOR agonist) induced the activation of DOR and the subsequent association

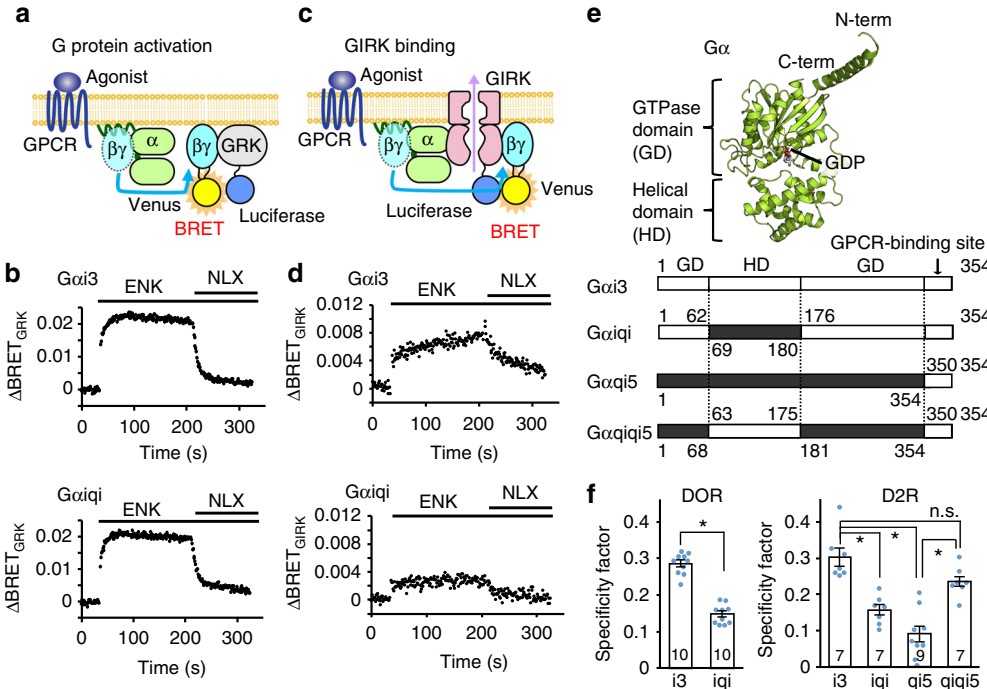

**Fig. 1** Measuring Gα specificity in Gβγ-GIRK binding. **a** Schematic representation of the BRET assay to measure G protein activation. Upon adding receptor agonists, the Venus-tagged Gβγ, the BRET acceptor, dissociates from Gα and then associates with the BRET donor GRK-Luc, leading to the increased BRET signal. **b** Representative traces of time-resolved BRET between Venus-Gβγ and GRK-Luc on cells expressing Gαi3 or Gαiqi along with DOR. The additions of the agonist Met-enkephalin (ENK) (10 μM) and the antagonist naloxone (NLX) (~83 μM) are indicated by bars. **c** Schematic representation of the BRET assay to measure GIRK-Gβγ binding. **d** Representative traces of time-resolved BRET between Venus-Gβγ and GIRK-Luc. **e** Top, the crystal structure of Gαi1 from Gαi1βγ (PDB ID: 1GP2)[33]. Bottom, topological representations of the Gα proteins used in this study. GD GTPase domain; HD helical domain. **f** The $\Delta BRET_{GIRK}/\Delta BRET_{GRK}$ ratios, named specificity factors, for each combination of GPCR and Gα. Data are means ± SEM. The number of measurements taken from independently transfected cell batches is indicated in the bar. *$p < 0.001$ by one-way ANOVA with post hoc Tukey–Kramer's test. Source data are provided as a Source Data File

between Venus-Gβγ and GRK-Luc, resulting in energy transfer between them to increase the BRET signal. The observed increase in the BRET signal ($\Delta BRET$) was reversibly decreased to the basal level by the addition of naloxone (DOR antagonist), showing that the observed BRET change reflects the binding between GRK and Gβγ, controlled by the DOR-mediated G protein signaling pathway. Similar changes in BRET signals were also observed in the cells expressing GIRK-Luc (Fig. 1c, d). These ligand-dependent changes in BRET were good indicators of the extents of GRK or GIRK activation. We confirmed that the effects of the basal activities on the measured BRET values were small, because further decreases in the BRET signal were not observed upon the application of inverse agonists of GPCR (Supplementary Table 1). The $\Delta BRET$ obtained using GIRK-Luc ($\Delta BRET_{GIRK}$) was normalized by using GRK-Luc ($\Delta BRET_{GRK}$), and we defined this $\Delta BRET_{GIRK}/\Delta BRET_{GRK}$ ratio as a "specificity factor" to quantitatively compare the preference of Gα for activating GIRK over GRK.

Rusinova and co-workers have previously reported that the helical domain of Gα confers the specificity for M2R-mediated GIRK activation[13]. Referring to this report, we compared the specificity factors when 3 different Gα proteins, Gαi3, Gαqi5, and Gαiqi, were used (Fig. 1e). Gαi3 belongs to the i/o family of Gα and is responsible for GIRK activation in biological processes; Gαqi5 refers to Gαq with the C-terminal 5 residues replaced by those of Gαi3 to couple with Gi/o-coupled GPCRs[26]; and Gαiqi is a chimeric protein consisting of the GTPase domain of Gαi3 (residues 1–62 and 176–354) and the helical domain of Gαq (residues 69–180)[13]. We confirmed that all of the Gα chimeric proteins used to calculate the specificity factor showed similar

$\Delta BRET_{GRK}$ values upon the addition of the GPCR agonists, indicating that they have comparable nucleotide binding properties and GPCR-coupling efficiencies (Supplementary Table 1–3). We also conducted competitive binding experiments, in which we monitored the decrease in the BRET signal caused by the displacement of Venus-Gβγ bound to GRK-Luc by increasing the amounts of Gα. In the cases of both Gαi3 and Gαiqi, the BRET signal decreased to a similar extent by increasing the amounts DNA encoding Gα, indicating that the replacement of the helical domain does not result in marked differences in the Gβγ-binding property in the inactive state (Supplementary Fig. 2A, B). Similar results were obtained when we expressed GIRK-Luc and monitored the $\Delta BRET_{GIRK}$ values (Supplementary Fig. 2C, D).

When we used the Gi/o-coupled receptor DOR, the specificity factors were 0.286 ± 0.009 and 0.148 ± 0.008 in cells expressing Gαi3 and Gαiqi, respectively ($n = 10$), and the specificity factor of Gαiqi was significantly smaller than that of Gαi3 ($p < 0.001$) (Fig. 1f). We could not observe either $\Delta BRET_{GRK}$ or $\Delta BRET_{GIRK}$ in cells expressing Gαqi5 ($\Delta BRET < 0.003$), indicating that Gαqi5 is not activated by DOR. We also compared the specificity factors using the Gi/o-coupled dopamine $D_2$ receptor (D2R), and obtained values of 0.303 ± 0.025, 0.091 ± 0.022, and 0.157 ± 0.015 for Gαi3, Gαqi5, and Gαiqi, respectively ($n = 7$–$9$), and the values obtained with Gαqi5 and Gαiqi were significantly smaller than that obtained with Gαi3 ($p < 0.001$) (Fig. 1f). To gain further insights into the role of the helical domain, we also prepared a chimeric Gα, Gαqiqi5, in which the helical domain of Gαqi5 (residues 69–180) is replaced with that from Gαi3 (residues 63–175), and found that the specificity factor of Gαqiqi5 (0.236 ± 0.016) was significantly larger than that of Gαqi5, and similar to

that of Gαi3. These results show that the Gβγ dissociated from Gαi3 or Gαqiqi5 binds to GIRK with significantly higher specificity than the Gβγ dissociated from Gαiqi and Gαqi5, even though the released Gβγ is identical, and this preference of Gα is commonly observed in both the DOR-mediated and D2R-mediated pathways. Since the difference among these Gα proteins exists mainly in the helical domain, our results strongly support the hypothesis that the helical domain of Gα is the major determinant that confers the Gα specificity in the activation of GIRK.

Together with the fact that the helical domain of Gαi/o(GTP) is involved in the binding to GIRK[21], we hypothesized that the Gαi/o specificity in the GIRK activation is attributable to the formation of a complex comprised of GIRK and Gαi/oβγ, in which the activation of GIRK is enhanced by the increased availability of Gβγ provided by the Gαi/oβγ that is colocalized with GIRK. This notion is further supported by the observation that the differences in the specificity factors between Gαi3 and Gαiqi markedly decreased in the cells expressing larger amounts of Gαβγ, where non-specific protein-protein encounters are facilitated and the formation of non-specific Gαiqiβγ–GIRK complexes tends to occur more frequently (Supplementary Fig. 3).

**NMR spectral changes of Gαi3βγ upon interaction with GIRK.** To determine whether the binding between Gαi/oβγ–GIRK actually occurs and contributes to the Gαi/o specificity, we set out to characterize the direct interaction between Gαi/oβγ and GIRK in an in vitro reconstituted system. In these analyses, we used a chimeric channel of GIRK1 (GIRK chimera), in which three-fourths of the transmembrane region were replaced with the pore of prokaryotic KirBac1.3[27]. The structure of the GIRK chimera is quite similar to that of the mammalian GIRK2[28], and the cytoplasmic region of the GIRK chimera is identical to that of the mammalian GIRK1. Since Gαi/oβγ is anchored to the cytoplasmic side of the membrane, the interaction with GIRK, if any, would occur on the cytoplasmic region of GIRK. Hence, the interaction between GIRK1 and Gαi/oβγ could be characterized by using the GIRK chimera. The GIRK chimera was reconstituted into phospholipid bilayer nanodiscs to mimic the physiologically relevant Gαi/oβγ–GIRK interaction that takes place on cell membrane[29]. We analyzed the interaction by using solution NMR techniques, which can characterize weak protein–protein interactions in physiological solution environments. In the analyses, we used a recombinant Gαi3βγ that lacks the lipid modification, which is partially localized to the lipid bilayer surface of the nanodiscs via an N-terminal polybasic region[30,31]. The experiments were conducted under physiologically-relevant ionic conditions (KCl = 150 mM), to suppress the non-specific binding mediated by this polybasic, positively charged regions.

We observed the NMR spectra of Gαi3βγ in the absence and presence of the GIRK chimera-nanodiscs to investigate whether Gαi3βγ interacts with the GIRK chimera and identify the regions that are affected upon the interaction. Due to the large molecular weights of Gαi3βγ (87 K) and the GIRK chimera-nanodisc (~380 K), we adopted selective methyl-labeling strategies and applied methyl-TROSY techniques[22]. We focused on observing the Gα subunit, since it confers the specificity, and prepared a selectively labeled {ul-[$^2$H, $^{15}$N]; Alaβ, Ileδ1, Leuδ, Valγ-[$^{13}$CH$_3$]} Gαi3 complexed with [non-labeled]βγ (Gαi3[ILVA]βγ). We observed the $^1$H–$^{13}$C HMQC spectrum of Gαi3[ILVA]βγ, and assigned the methyl signals based on the nuclear Overhauser effect spectroscopy and mutagenesis experiments (Supplementary Fig. 4). By comparing the HMQC spectrum with that of Gαi3 alone, which we previously reported[32], we confirmed the formation of the

Gαi3βγ complex that is consistent with the reported crystal structure[33] (Supplementary Fig. 5). Upon the addition of 2 equivalents of the GIRK chimera-nanodiscs to Gαi3[ILVA]βγ, most of the signals exhibited intensity reductions with relative intensities lower than 0.9, and the signals from L5δ1, L5δ2, A12β, V13γ1, V13γ2, A30β, A31β (N-terminal helix), L36δ1, L36δ2, L37δ1, L37δ2 (β1 strand), A41β (β1-α1 loop), I127δ1(αC helix), L148δ1(αD-αE loop), L159δ2 (αE helix), V218γ2 (α2-β4 loop), I221δ1(β4 strand), L232δ2, L234δ2 (β4-α3 loop), L249δ1, L249δ2, I253δ1 (α3 helix), I264δ1, I265δ1 (β5 strand), I278δ1, L283δ1 (αG-α4 loop), and L348δ1 (C-terminus) exhibited further reduced intensities lower than 0.8 (Fig. 2a–c), while the observed chemical shift changes were very small (<0.01 ppm). When we added the empty nanodiscs, we did not observe significant intensity reductions, showing that the observed intensity reductions upon the addition of the GIRK chimera-nanodiscs are mainly triggered by the specific binding of Gαi3[ILVA]βγ to the GIRK chimera (Fig. 2c). The overall intensity reductions are caused by slower tumbling, due to an increased average molecular weight, indicating that a fraction of Gαi3[ILVA]βγ forms a complex with the GIRK chimera-nanodiscs. The further intensity reductions are caused by differential line broadening, which results from the chemical shift changes in an intermediate-to-fast exchange regime between the free and the bound states, and/or the effect of the anisotropic tumbling induced by the binding, although the effect of the anisotropic tumbling was estimated to be relatively small for membrane proteins in nanodiscs[34]. Since the total molecular weight of the complex is quite large for NMR observation (>400 K) and the interaction is relatively weak, the binding effects are mainly observed as reductions in the signal intensities, caused by the differential line broadening[35], in a similar manner to the interaction between the cytoplasmic region of GIRK and Gαi3(GTP)[21]. Assuming that the overall intensity reduction (~0.1) reflects the apparent increase in the molecular weight as a function of the bound population, we estimated the apparent $K_d$ to be larger than 200 μM. The residues with significant intensity reductions were located on the N-terminal and C-terminal regions, the Gβγ-binding site within the GTPase domain, and the helical domain of Gαi3 (Fig. 2d), suggesting that these regions exhibit chemical shift differences caused by the direct contact with the GIRK chimera-nanodiscs, and/or by the conformational changes that occur upon the interaction. This estimation of the $K_d$ value is also consistent with the site-specific intensity reductions that were as large as 0.3, if we assume that the on-rate is diffusion limited ($k_{on}$ ~10$^7$ M$^{-1}$ s$^{-1}$) and the $^1$H chemical shift difference between the free-state and bound-state is around 0.05–0.1 ppm. Since Gαi3 is anchored to the membrane at its N-terminus under physiological conditions, the spectral changes observed in the N-terminal region and the neighboring C-terminal region may reflect the binding of Gαi3βγ to the membrane lipids of the nanodiscs. As the N-terminal region of Gαi3 simultaneously interacts with Gβγ, the Gβγ-binding site might be slightly affected upon membrane-anchoring via the N-terminal region, resulting in the intensity reduction observed on the Gβγ-binding site. The helical domain of Gαi3 is distant from the membrane-binding site, so the chemical shift differences in this domain might be caused by interactions with the GIRK chimera. Together, these spectral changes strongly suggest the direct interaction of Gαi3βγ with the GIRK chimera-nanodiscs.

We performed the same experiment using Gαiqi[ILVA]βγ (Supplementary Fig. 6). In contrast to Gαi3[ILVA]βγ, we did not observe an overall intensity reduction upon the addition of the GIRK chimera-nanodiscs, and no signals exhibited reduced intensities lower than 0.8 (Fig. 2b, c). These results demonstrated that Gαiqiβγ has significantly lower affinity for the GIRK chimera-nanodiscs than Gαi3βγ. Together with the results of

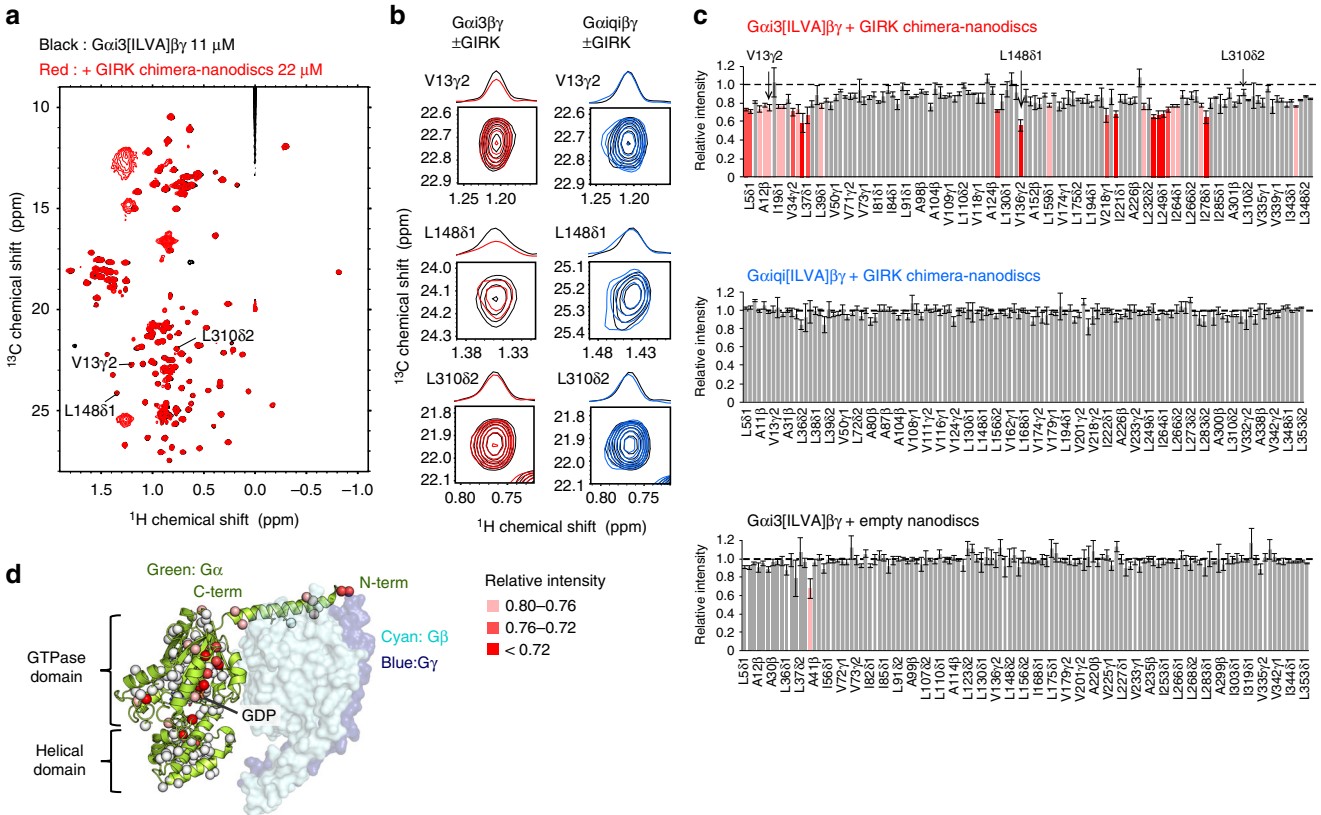

**Fig. 2** NMR spectral changes of Gαi3[ILVA]βγ induced by the GIRK chimera-nanodiscs. **a** Overlay of the ¹H–¹³C HMQC spectra of Gαi3[ILVA]βγ in the presence (red) and absence (black) of 2 eq. of the GIRK chimera-nanodiscs. **b** Close-up views and cross-sections of V13γ2, L148δ1, and L310δ2 as representative signals. For comparison, the corresponding signals of Gαiqi[ILVA]βγ in the presence (blue) and absence (black) of the GIRK chimera-nanodiscs are shown on the right. **c** Plots of relative intensities of Gαi3[ILVA]βγ (top) and Gαiqi[ILVA]βγ (middle) in the presence of 2 eq. of the GIRK chimera-nanodiscs. A plot of the relative intensities of Gαi3[ILVA]βγ in the presence of empty nanodiscs is also shown (bottom). The error bars are calculated based on the signal-to-noise ratios. Methyl groups with relative intensities lower than 0.80 are colored according to their values. **d** Mapping of the methyl groups of Gαi3βγ with significant intensity reductions on the structure of Gαi1βγ (PDB ID: 1GP2)[33]. Methyl groups are shown as spheres and colored according to their relative intensity values. Source data are provided as a Source Data File

our cell-based BRET experiments, in which Gαiqi did not efficiently provoke the activation of GIRK, we concluded that Gαi3βγ directly interacts with GIRK through its helical domain, and the interaction is responsible for the Gαi/o-specific GIRK activation.

**Interacting sites of Gαi3βγ–GIRK complex revealed by PRE.** In order to gain insight into the structure of the Gαi/oβγ–GIRK complex, we conducted paramagnetic relaxation enhancement (PRE) experiments. PRE arises from the magnetic dipolar interaction between a nuclear spin and an unpaired electron of the paramagnetic center, resulting in line-broadening of the NMR signal of the nuclear spin, depending on the distance from the paramagnetic center. The distance information within the complex can be obtained from the PREs observed in the free state signals, since PREs are transferred from the transiently-formed bound state to the free state in the fast-exchanging system[36,37].

To collect the distance information, we site-specifically labeled the GIRK chimera with a spin-labeling reagent, 4-maleimido-2,2,6,6-tetramethylpiperidine-1-oxyl (TEMPO), which can be covalently ligated to cysteine side chains, and measured the PREs observed on Gαi3[ILVA]βγ. We first constructed a mutant of the GIRK chimera (C53S/C310T), which has no reactive cysteine residue. Using this mutant as a template, Q344, V351, and L366 were separately replaced with cysteine for the site-specific spin labeling of the GIRK chimera. These three residues are distributed

across the entire cytoplasmic domain of the GIRK chimera, and thus they would allow us to identify the relative position of Gαβγ to the GIRK chimera. Q344 and V351 are located on the βM-βN loop and the βN-C-terminal helix loop, respectively, which are both within the highly structured β-strand region, while L366 is located on the C-terminal helix of the GIRK chimera (Fig. 3a). The PRE contributions to the transverse relaxation rates, $\Gamma_2$, were measured using the signal intensities of Gαi3[ILVA]βγ in the presence of the spin-labeled GIRK chimera-nanodiscs, before and after the paramagnetic center of 4-maleimido-TEMPO was reduced by ascorbate.

The results are summarized in Fig. 3. Significant $\Gamma_2$ values over $5\,s^{-1}$ caused by L366C-TEMPO were observed for the signals of I85δ1, L130δ1, L130δ2, and V136γ2 of Gαi3. These methyl groups are clustered around the αA and αB helices in the helical domain of Gαi3 (Fig. 3b). A few signals (V126γ2 and A299β) exhibited $\Gamma_2$ larger than $5\,s^{-1}$ caused by Q344C-TEMPO and V351C-TEMPO, and these methyl groups are not clustered on the structure (Fig. 3c). Based on these PRE patterns, we concluded that the C-terminal helix of GIRK, where L366 is located, is proximate to the helical domain of Gαi3 in the Gαi3βγ–GIRK complex, while the β-strand regions of the cytoplasmic region of GIRK do not form stable interactions with Gαi3βγ.

**Constructing a model structure of the Gαi3βγ-GIRK complex.** We sought to visualize the structure of the Gαi3βγ-GIRK

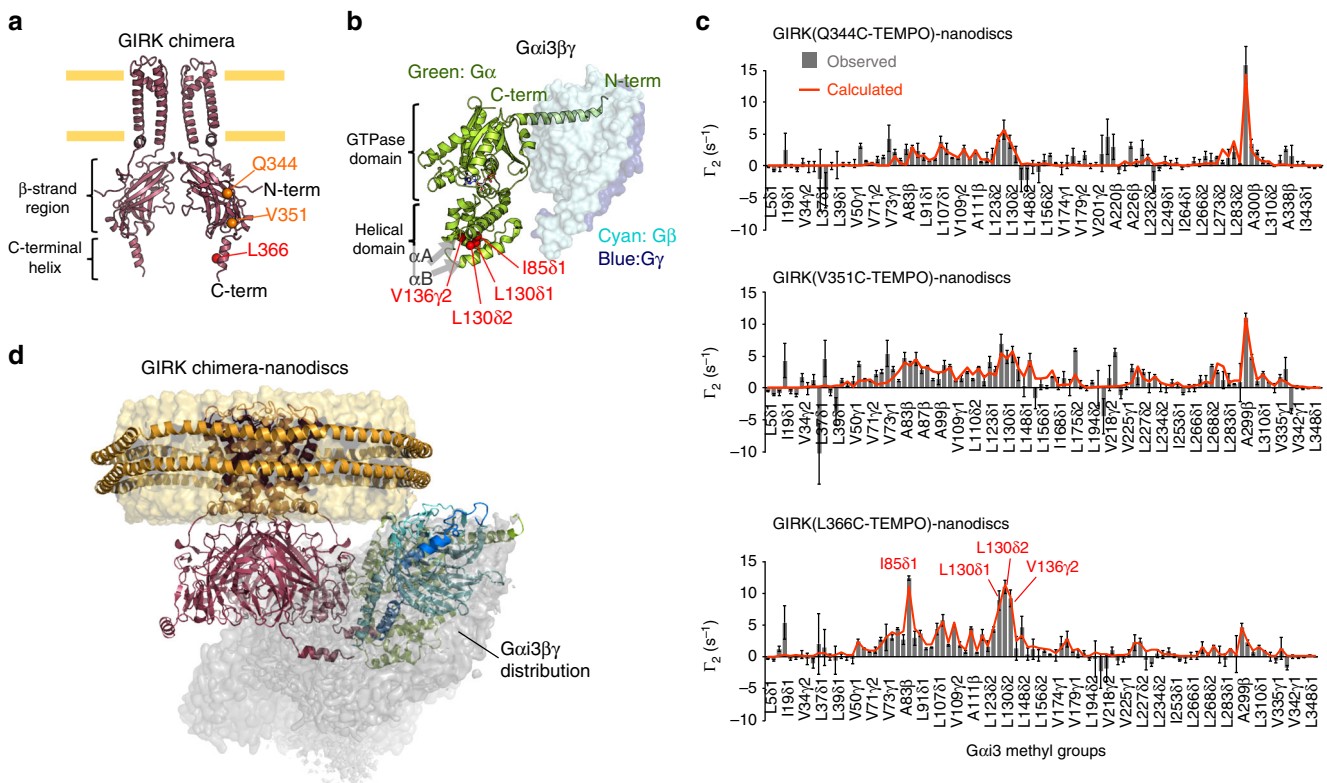

**Fig. 3** Observed and calculated PREs between the GIRK chimera and Gαi3βγ. **a** The locations of Q344, V351, and L366 are shown on the structure of the GIRK chimera (modified from PDB ID: 2QKS; see methods)[27]. **b** The methyl groups with significant PREs ($\Gamma_2^{obs} > 5\ s^{-1}$) in the experiment using L366C-TEMPO are shown as red spheres on the crystal structure of Gαi1βγ (PDB ID: 1GP2)[33]. **c** $\Gamma_2$ rates observed in the PRE experiments, $\Gamma_2^{obs}$ (bars), and those back-calculated from the ensemble structure, $\Gamma_2^{calc}$ (orange lines) are shown. The signals with significant PREs ($\Gamma_2^{obs} > 5\ s^{-1}$) from L366C-TEMPO are labeled. The error bars are calculated based on the signal-to-noise ratios. **d** The distribution of Gαi3βγ relative to the GIRK chimera in the obtained ensemble structure. An atomic probability density map is displayed as a surface representation at the contour level of $\rho = 0.05$. One orientation in which Gαi3βγ is directed toward the membrane is shown in the ribbon diagram. Source data are provided as a Source Data File

complex by structural calculation using the observed PREs as distance restraints. However, our initial attempt to obtain a single complex structure that simultaneously satisfies the PRE patterns from Q344C-TEMPO, V351C-TEMPO, and L366C-TEMPO failed, as indicated by the relatively large Q-factor[38] of 0.71, even in the best fit result. This result indicates that the relative orientation between Gαi3βγ and GIRK in the complex is inherently flexible, and we must use an ensemble of structures to explain the observed PREs. Therefore, we calculated an ensemble of multiple structures that explains the experimental PRE data. The calculations were performed in two steps: First, we docked the C-terminal helix of GIRK to Gαi3, based on the major PREs obtained from L366C-TEMPO. Second, to recapitulate the relatively minor PRE patterns from Q344C-TEMPO and V351C-TEMPO, we generated 30,000 possible structures considering the conformational flexibility of the GIRK region (residues 352 to 357) connecting the β-strand region with the C-terminal helix, and then optimized the weight of each structure. The calculated PREs from the weighted ensemble of the selected 1000 structures agreed with the experimental PREs from Q344C-TEMPO, V351C-TEMPO, and L366C-TEMPO, with an overall Q-factor of 0.421 (Fig. 3c orange lines and Supplementary Fig. 7), indicating that the ensemble illustrates the interaction mode between the GIRK chimera and Gαi3βγ under the experimental conditions.

To visualize the spatial distribution of Gαi3βγ relative to the GIRK chimera, we calculated the weighted atomic probability density[39] (Fig. 3d). In the obtained ensemble consisting of 1000 orientations, the location of Gαi3βγ ranged from beside the

membrane to below the β-strand region of GIRK, while retaining the interaction between the C-terminal helix of GIRK and the helical domain of Gαi3. Gαi3βγ distribution in the ensemble was different from the randomly generated distributions (Supplementary Fig. 8), so the calculated ensemble is likely to represent the orientations of Gαi3βγ while interacting with the GIRK chimera-nanodiscs. Notably, the ensemble included several orientations in which the N-terminus of Gαi3 and the C-terminus of Gγ are directed toward the membrane (Fig. 4a and Supplementary Fig. 9). These orientations are consistent with the membrane anchoring of lipidated Gαβγ in vivo, and thus we concluded that these orientations represent the physiological interaction mode of Gαi/oβγ–GIRK (Fig. 4a and Supplementary Fig. 9). In the ensemble structures, the C-terminal helix of GIRK and the αA helix in the helical domain of Gαi3 form a major binding surface (Fig. 4a). While the amino acid sequences of the helical domain, especially those of the αA, αB, and αC helices, are less conserved among G protein families (Fig. 4b), our structural model suggested that the αA helix is the key structural element that mediates the formation of the Gαi/oβγ–GIRK complex, and hence determines the Gαi/o specificity in the activation of GIRK. To verify this model, we conducted a structure-guided mutational analysis. We constructed a chimeric Gαi3 in which the αA helix (residues 71–90) was replaced by that of Gαq (Gαi3-q(αA)), and tested the effect on the specificity factor by BRET assays. For comparison, we also used chimeras in which other structural elements, the αB (residues 100–110) and αE (residues 151–163) helices, were replaced with those of Gαq (Gαi3-q(αB) and Gαi3-q

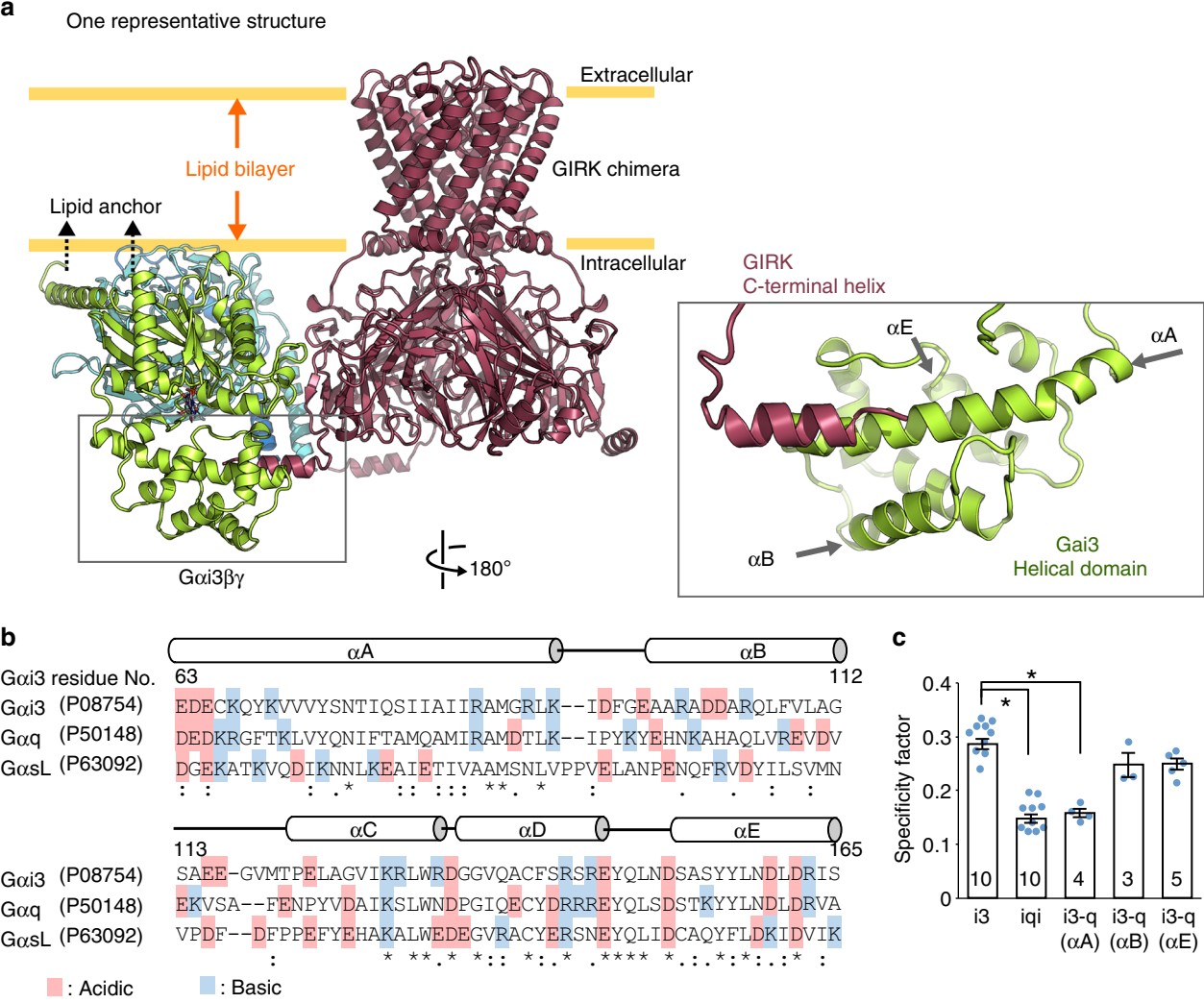

**Fig. 4** The binding mode of Gαi3βγ and GIRK. **a** One of the representative structures of the transient Gαi3βγ–GIRK chimera complex, in which the membrane-anchoring sites of Gαi3βγ are directed toward the membrane. A close-up view of the interacting site is shown on the right. **b** The sequence alignment of the helical domains of Gαi3, q, and s, based on the alignment by Flock et al.[9]. UniProt accession numbers are shown in parentheses. Acidic and basic amino acids are colored red and blue, respectively. Secondary structures are shown on the top. **c** The specificity factors ($\Delta BRET_{GIRK}/\Delta BRET_{GRK}$ ratios) for Gα mutants measured by BRET assays. Data are means ± SEM. The number of measurements taken from independently transfected cell batches is indicated in the bar. *$p < 0.001$ by one-way ANOVA with post hoc Tukey-Kramer's test. Source data are provided as a Source Data File

(αE)). The specificity factors of these chimeras are shown in Fig. 4c. The specificity factor for Gαi3-q(αA) was $0.159 \pm 0.008$ ($n = 4$), which was significantly smaller than that of the wild-type Gαi3 ($0.286 \pm 0.009$, $p < 0.001$). In contrast, the specificity factors of Gαi3-q(αB) and Gαi3-q(αE) were $0.248 \pm 0.023$ ($n = 3$, $p = 0.041$ vs. Gαi3) and $0.249 \pm 0.011$ ($n = 5$, $p = 0.022$ vs. Gαi3) respectively, which were statistically not significantly different from that of the wild-type Gαi3. We also conducted NMR experiments to observe Gαi3-q(αA)[ILVA]βγ, and significant intensity reductions were not found upon the addition of the GIRK chimera-nanodiscs, indicating that the specific binding to the GIRK chimera was diminished in Gαi3-q(αA)[ILVA]βγ (Supplementary Fig. 10). From these results, we concluded that the αA helix is the key structural element of Gαi/o that couples specifically with GIRK.

## Discussion

Our cell-based BRET experiments revealed that the activation of GIRK is invoked by Gα which possesses the helical domain of the Gαi/o family regardless of the GPCR type, indicating that the helical domain of Gα is the major determinant of the family specificity in GIRK activation (Fig. 1). Our NMR experiments using the purified Gαi3βγ and the GIRK chimera proved that the helical domain of Gα contributes to the formation of the Gαi3βγ-GIRK complex (Fig. 2), by directly-binding to the C-terminal helix of GIRK (Figs. 3, 4). These two lines of evidence indicate that the formation of the Gαi/oβγ–GIRK complex is responsible for the Gαi/o-specific GIRK activation, and the interaction between the helical domain of Gα and the C-terminal helix of GIRK mediates the complex formation. Our results clearly demonstrate that the interaction formed between GIRK-Gα, rather than that between GIRK-GPCR or GIRK-Gβγ, plays a crucial role in the formation of the complex, and provide detailed structural insights into the mechanism by which Gβγ in the complex immediately interacts with GIRK upon the activation of Gαi/o, thereby achieving the Gαi/o specificity in GIRK activation[19,40,41].

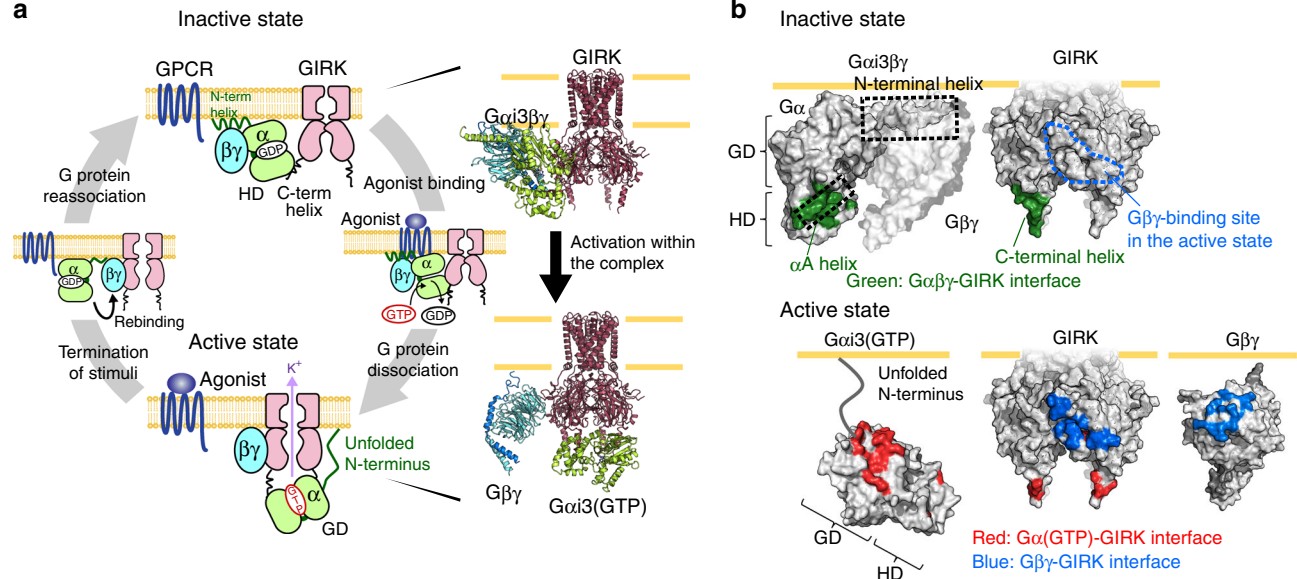

**Fig. 5** Persistent Gi/o protein-GIRK coupling throughout the activation cycle. **a** (Left) Schematic representation of colocalizing GIRK and Gi/o proteins in the activation cycle. (Right) Ribbon diagrams of a representative structure of the Gαi3βγ–GIRK complex in the ensemble model and the model structure of the Gαi3(GTP)–GIRK–Gβγ complex, using the crystal structures of the GIRK chimera (PDB ID: 2QKS_1)[27], Gαi1βγ (PDB ID: 1GP2)[33], Gαi1(GTP) (PDB ID: 1GIA)[63], and Gβγ (from PDB ID: 4KFM)[42]. The model of the Gαi3(GTP)–GIRK–Gβγ complex was built by integrating the model of the Gαi3(GTP)–GIRK complex[21] with the crystal structure of the GIRK–Gβγ complex[42]. **b** Components of the complex in the inactive (top) and active (bottom) states are shown as surfaces. Residues on the Gαi3βγ–GIRK binding interface (residues within 5.5 Å distance) are colored green. Residues on the Gαi3(GTP)–GIRK binding surfaces identified by transferred cross-saturation experiments[21], and the Gβγ–GIRK binding surfaces[42,43] are colored red and blue, respectively. GD GTPase domain, HD helical domain

The helical domain of Gα is reportedly important for coupling to GIRK[13], but the key residues responsible for the coupling could not be identified by simply comparing their amino acid sequences, due to the low sequence conservation (Fig. 4b). Our model structure of the Gαi3βγ–GIRK complex demonstrated that the αA helix of Gαi3 directly interacts with the C-terminal helix of GIRK, and the importance of this region for the functional coupling with GIRK was further supported by the structure-guided mutational analyses, using the Gα chimera of the αA helix (Fig. 4c and Supplementary Fig. 10). Closer inspection of the complex structure also revealed the charge complementarity between the positively charged αA helix of Gαi3 and the negatively charged C-terminal helix of GIRK (Supplementary Fig. 11). Since the charge distributions around the αA helix significantly differ among the Gα families, this charge complementarity is likely to promote the formation of the Gαi/oβγ–GIRK complex in a family-specific manner. We note that the interaction between the αA helix of Gαi3 and the C-terminal region of GIRK was similarly observed in a Gαi/o(GTP)–GIRK complex formed during the activation of GIRK, which we previously characterized by intermolecular PRE and transferred cross-saturation experiments[21]. This observation indicates that the helical domain of Gαi3, where the αA helix is located, has an inherent affinity for GIRK, and that the interaction is the driving force for the interaction between Gαi3βγ and GIRK.

Our comparisons of the binding modes of Gβγ–GIRK, Gαi/o(GTP)–GIRK, and Gαi/oβγ–GIRK led us to propose the structural basis underlying the regulation of the GIRK gating throughout the G protein activation cycle, which enables the specific and efficient responses to the extracellular stimuli received by GPCRs. In the inactive state, G proteins are in the GDP-bound heterotrimeric Gαβγ form and bind to GIRK. Within the complex, Gαi/oβγ and GIRK are tethered to each other, mainly via the helical domain of Gαi/o and the C-terminal helix of GIRK (Fig. 5a). Upon the activation of GPCRs by agonist

binding, the GPCRs catalyze the GDP–GTP exchange reaction on Gαi/o, causing Gαi/oβγ to dissociate into Gαi/o(GTP) and Gβγ. This allows the GIRK-binding site on Gβγ, which is covered by Gα in Gαi/oβγ, to become available. Since the Gαi/oβγ-binding site on GIRK does not overlap with the Gβγ-binding site[42,43] (Fig. 5b), the dissociated Gβγ can rapidly bind to the nearby GIRK in the pre-formed complex, thus efficiently inducing the opening of GIRK. Only the i/o family of Gαβγ can form the complex with GIRK, which enables the i/o family to specifically participate in the GIRK activation. Gαi/o(GTP) binds to the C-terminal helix of activated GIRK through its GTPase domain, as we previously reported[21]. Following the termination of extracellular stimuli and the hydrolysis of GTP to GDP by its intrinsic GTPase activity, Gαi/o(GDP) rebinds to Gβγ with nanomolar affinity[44], competitively removing Gβγ from GIRK, which leads to its immediate closure. These structural models suggest that the stepwise changes in the interaction modes between GIRK and G proteins enable the specific and efficient regulation of GIRK in response to extracellular stimuli, which would play critical roles in the robust and selective signal transductions in the heart and neural systems (Please also see Supplementary Discussion). We expect that the regulation of other effector proteins, such as adenylate cyclase, is not affected by the formation of the Gαi/oβγ–GIRK complex, because these effectors do not co-localize with GIRK and can be independently regulated by G proteins that do not participate in the Gαi/oβγ–GIRK complex.

Several lines of evidence have indicated that the signaling complex also contains GPCR, along with G protein and GIRK[12–14]. The GPCR-binding site on Gαβγ is the α5 helix in the GTPase domain of Gα[15], and since this region is not involved in the interaction with GIRK, GPCR can also bind to Gαi/oβγ in the proposed complex structure (Supplementary Fig. 12). These observations suggest that the Gαi/oβγ–GIRK complex revealed here represents part of an even larger complex including GPCR, and the formation of this large signaling complex underlies the

Gα family specific activation of GIRK. This model is in contrast to the model recently proposed by Touhara and MacKinnon[16], in which the specificity of GIRK signaling is attributed to the difference in the association rate of G protein with GPCR, rather than by a specific binding of Gαβγ to GIRK. Although the origin of the family specificity is different between these two models, we note that these two models commonly assume that the local concentrations of GPCR, GIRK, and G proteins are maintained at high levels to achieve sufficient concentration of Gβγ to invoke the activation of GIRK upon the activation of GPCR. We also note that the mechanisms underlying the Gα family specificity may be different between Gαi/o versus Gαs investigated by Touhara and MacKinnon, and Gαi/o versus Gαq investigated in our study.

In summary, we investigated the molecular mechanism underlying the Gαi/o-specific activation of GIRK. By combining cell-based BRET experiments and NMR analyses, we showed that the helical domain of Gα is a major determinant in the formation of the Gαi/oβγ–GIRK complex, and the formation of the complex enables the family-specific activation of GIRK. Based on the distance restraints obtained from the PRE experiments, we constructed the model structure of the Gαi/oβγ–GIRK complex, in which the αA helix of the helical domain of Gα forms the major binding surface for GIRK. This complex provides the molecular basis underlying the specific and efficient regulation of the GIRK gating throughout the G protein cycle. Our results demonstrate that the transient protein-protein interactions that occur on the membrane surface play critical roles in defining the signaling pathways in physiological contexts. Since the interactions between Gαβγ and other effector proteins, such as adenylate cyclase, have been also proposed to be important for their physiological functions[45], our results will further facilitate the comprehension of intracellular signaling networks, and also highlight the importance of NMR techniques that can characterize transient interactions involving biologically important signaling molecules.

## Methods

**BRET assays.** The DNA fragments encoding GIRK1, GIRK2, G protein-coupled receptor kinase 3 (GRK3), delta opioid receptor (DOR), dopamine $D_2$ receptor (D2R), Gαi3, Gαq, Gβ$_1$, and Gγ$_2$ were amplified from human whole brain cDNA (Clontech) using primers listed in Supplementary Table 4. The chimeric Gα proteins Gαiqi[13], Gαi3-q(αA), Gαi3-q(αB), and Gαi3-q(αE) were constructed by using Gαi3 as a template and replacing the sequences of residues 63–175, 71–90, 100–110, and 151–163, respectively, with those of Gαq, using the In-Fusion HD Cloning Kit (TaKaRa). Gαqi5 was constructed by replacing the C-terminal 5 residues of Gαq with those of Gαi3[26]. The DNA fragments encoding NanoLuc (NLuc) were amplified from the pNL1.1 plasmid (Promega) using primers listed in Supplementary Table 4. The DNA fragments encoding Venus and the S1 catalytic subunit of pertussis toxin (PTX S1) were synthesized by GeneArt Strings™ DNA Fragments (Thermo Fisher Scientific). Venus 156–239-Gβ$_1$ and Venus 1–155-Gγ$_2$, which dimerize to form Venus-tagged Gβγ (Venus-Gβγ) were constructed by fusing the fragment of Venus to a GGSGGS linker and the N-terminus of Gβ$_1$ or Gγ$_2$[23,24]. NLuc-tagged GRK3 construct (masGRK3ct-Luc) was made by fusing the residues 495–688 of GRK3 preceded by a myristic acid attached peptide to a GGGS linker and the N-terminus of NLuc[23,24]. All DNA fragments were inserted into the pcDNA 3.1/Zeo (+) expression vector (Invitrogen). Site-specific mutations were introduced using a QuikChange® Site-Directed Mutagenesis Kit (Agilent technologies).

HEK293T cells were purchased from ECACC [293T (ECACC 12022001)]. The cells were cultured in Dulbecco's modified Eagle's medium (DMEM) supplemented with 10% FBS (Biowest), 1× GlutaMAX (Gibco), sodium pyruvate (Gibco), and Penicillin-Streptomycin (Gibco) at 37 °C in a 5% CO$_2$ atmosphere. For transfection, the cells were seeded on 6-well plates (Corning) at a density of $6 \times 10^5$ cells/well. On the next day, the cells were transfected with DNA plasmids using Lipofectamine 3000 (Invitrogen).

In the BRET assays, we co-expressed PTX S1 to avoid the activation of endogenous Gαi/o, and used PTX-insensitive Gα (C351A) mutants to selectively observe the activation of the expressed Gα. The following combinations of DNA plasmids were used (Tables 1–4).

About 24 h after transfection, cells were detached by PBS containing 5 mM EDTA, harvested by centrifugation at $400 \times g$ for 3 min, and resuspended in 1 mL

### Table 1 DNA plasmids used for transfection including DOR

| Construct | DNA (ng/well) |
|---|---|
| DOR | 48 |
| PTX S1 | 190 |
| Gα PTX-insensitive C351A mutant | 380 |
| Venus 156–239-Gβ$_1$ | 24 |
| Venus 1–155-Gγ$_2$ | 24 |
| masGRK3ct-Luc or GIRK1/GIRK2-Luc | 18 or 18/18 |

### Table 2 DNA plasmids used for transfection including D2R

| Construct | DNA (ng/well) |
|---|---|
| D2R | 48 |
| PTX S1 | 190 |
| Gα i3, iqi, qi5, or qiqi5 (C351A) | 285, 190, 380, or 285 |
| Venus 156–239-Gβ$_1$ | 24 |
| Venus 1–155-Gγ$_2$ | 24 |
| masGRK3ct-Luc or GIRK1 / GIRK2-Luc | 18 or 18/18 |

### Table 3 DNA plasmids used for transfection to vary the amount of Gαβγ

| Construct | DNA (ng/well) | | |
|---|---|---|---|
| DOR | 48 | 48 | 48 |
| PTX S1 | 190 | 190 | 190 |
| Gα i3 or iqi (C351A) | 380 | 570 | 950 |
| Venus 156–239-Gβ$_1$ | 24 | 95 | 190 |
| Venus 1–155-Gγ$_2$ | 24 | 95 | 190 |
| masGRK3ct-Luc or GIRK1/GIRK2-Luc | 18 or 18/18 | 24 or 24/24 | 47.5 or 47.5/47.5 |

### Table 4 DNA plasmids used for transfection for Gα titration

| Construct | DNA (ng/well) | |
|---|---|---|
| DOR | 48 | 48 |
| PTX S1 | 190 | 190 |
| Gα i3 or iqi (C351A) | 0 | 95, 190, 285, 380, 570 |
| Venus 156–239-Gβ$_1$ | 0, 24 | 24 |
| Venus 1–155-Gγ$_2$ | 0, 24 | 24 |
| masGRK3ct-Luc or GIRK1/GIRK2-Luc | 18 or 18/18 | 18 or 18/18 |

of BRET buffer (PBS containing 0.5 mM MgCl$_2$ and 0.1% (w/v) glucose). Each well of a white 96-well plate (Perkin Elmer OptiPlate-96) was loaded with 25 μl of cell suspension (containing 50,000–100,000 cells), 75 μL of BRET buffer, and 25 μL of a 5× solution of Nano-Glo® Luciferase Assay Substrate (Promega). Venus (535 nm) and NLuc (460 nm) emissions were measured on a 2030 ARVO X5 plate reader (Perkin Elmer). Each sample was measured in triplicate, and the average value was used. The BRET ratios were determined by calculating (emission of Venus)/ (emission of NLuc). Agonists of GPCR were added at a final concentration of 10 μM (or 1 μM for Met-enkephalin before addition of ICI-174,864), followed by a 10-fold molar excess of antagonists or inverse agonists. Data were recorded 3 min after addition of ligands.

To assess the expression of Venus-Gβγ, cells were seeded on a 35-mm glass-based dish (IWAKI) and transfected as described above. At 24 h after transfection, the cells were fixed with 4% paraformaldehyde in PBS for 20 min and washed once with PBS. To assess the expression of GIRK1/GIRK2-Luc, double immunostaining was performed. All antibodies were purchased from Abcam. The cells were transfected and fixed as described above, and permeabilized with 0.2% Triton X-100 in PBS for 5 min. The cells were washed twice with PBS, incubated in PBS

containing 1% BSA and 0.05% Tween 20 for 30 min, and then incubated with rabbit anti-GIRK1 and goat anti-GIRK2 for 1 h. After three washes with PBS, the corresponding second antibodies, anti-rabbit antibody-Alexa Fluor 488 and anti-goat antibody-Alexa Fluor 647, were added. After a 60-min incubation, the cells were washed three times and observed by microscopy. Confocal microscopy was performed using an FV10i microscope (Olympus).

**Protein expression and purification.** The Gαi3 (residues 1–354) protein, expressed with an N-terminal His10-tag and an HRV 3C protease cleavage site, was produced in *Escherichia coli* BL21-CodonPlus (DE3)-RP cells. For the selective $^{13}CH_3$-labeling of methyl groups, the *E. coli* cells were grown in deuterated M9 media, and 50 mg L$^{-1}$ of [3,3-$^2H_2$, 4-$^{13}C$] α-ketobutyric acid (for Ileδ1), 100 mg L$^{-1}$ of [3,4,4,4-$^2H_2$, 4-$^{13}C$] α-ketoisovaleric acid (for Leu/Val-[$^{13}CH_3$, $^{12}CD_3$] labeling), 120 mg L$^{-1}$ of [3-$^2H_2$, 4,4-$^{13}C_2$] α-ketoisovaleric acid (for Leu/Val-[$^{13}CH_3$, $^{13}CH_3$] labeling), 300 mg L$^{-1}$ of [2-$^{13}C$, 4,4,4-$^2H_3$] acetolactate (for Leu/Val $^{proS}$-[$^{13}CH_3$] labeling), or 200 mg L$^{-1}$ of [2-$^2H$, 3-$^{13}C$] alanine (for Alaβ) with 2.5 g L$^{-1}$ of [$^2H_6$] succinate were added, 30 min prior to the induction[46,47]. The Gαi3 protein was purified by chromatography on HIS-Select Nickel Affinity Gel (Sigma-Aldrich), cleavage of the His-tags with HRV-3C protease (Novagen), and removal of the cleaved His-tags and the protease on HIS-Select Nickel Affinity Gel[32,48]. All mutant and chimeric Gα proteins were prepared in the same way as the wild-type Gαi3.

The expression and purification of Gβγ were performed as follows[43]. The human Gβ₁ and N-terminally His6-tagged human Gγ₂ (C68S) were subcloned into the pFastBac Dual vector (Invitrogen). Gγ₂ (C68S) does not undergo lipid modification. The recombinant baculovirus was amplified in Sf9 cells using the Bac-to-Bac Baculovirus Expression System (Invitrogen). The Gβγ dimer protein was expressed by infecting ExpreSF + insect cells (Protein Sciences) with the baculovirus at a multiplicity of infection (MOI) of 4. After an incubation at 27 °C for 48 h, the cells were harvested and resuspended in 50 mL of buffer (20 mM Tris-HCl (pH 8.0), 150 mM NaCl, 1 mM MgCl₂, 1 mM dithiothreitol (DTT), containing 1× Protease Inhibitor Cocktail (EDTA-free) (Nacalai Tesque)) per 1 L culture. The purification procedure was performed at 4 °C. The cells were disrupted by nitrogen cavitation (Parr Instrument) or sonication. The cell lysate was then centrifuged, and the supernatant was purified using HIS-Select Nickel Affinity Gel, followed by further purification on DEAE Sepharose (GE Healthcare) and HisTrap HP (GE Healthcare) columns.

To obtain the heterotrimeric Gαβγ protein, the purified Gα and Gβγ proteins were mixed with a slight excess of Gβγ, and Gαβγ was isolated by gel filtration on Superdex 200 GL 10/300 or HiLoad Superdex 200 prep grade 26/600 columns (GE Healthcare). Gαi3βγ used in the structural analyses lacks the lipid modification, because we found that lipidated G proteins tend to aggregate during sample preparation. Nevertheless, we assumed that the recombinant Gαi3βγ is partially localized to the lipid bilayer surface of the nanodisc, since Gαi3 retains an N-terminal polybasic region, which interacts with acidic lipids and promotes membrane-targeting in analogy to other G proteins[30,31].

The KirBac1.3-GIRK1 chimeric protein (GIRK chimera)[27], consisting of mouse GIRK1 residues 41–386, in which residues 83–177 are replaced with residues 62–141 of *Burkholderia xenovorans* KirBac1.3, including an N-terminal His10-tag and an HRV-3C protease recognition site, was expressed in *Escherichia coli* C43 (DE3) cells (Lucigen). The GIRK chimera protein was solubilized in 20 mM of *n*-dodecyl-β-D-maltoside (Dojindo) and purified by chromatography on HIS-Select Nickel Affinity Gel. For nanodisc reconstitution, a lipid mixture comprising 70% 1-palmitoyl-2-oleoyl-phosphatidylcholine, 25% 1-palmitoyl-2-oleoyl-phosphatidylglycerol, and 5% L-α-phosphatidylinositol-4,5-bisphosphate (Brain, Porcine) (w/w) (Avanti Polar Lipid) was desiccated and dissolved in 50 mM sodium cholate. We used nanodiscs composed of MSP1E3 with an approximate diameter of 120 Å[49], which is sufficiently large to accommodate the complex of Gαi/oβγ with about a 40 Å length at the lipid-binding N-terminal region, and the GIRK chimera with a diameter of 45 Å at the transmembrane region. The MSP1E3 protein was prepared by chromatography on HIS-Select Nickel Affinity Gel (Sigma-Aldrich), cleavage of the His-tags with TEV protease, and removal of the cleaved His-tags and the protease on HIS-Select Nickel Affinity Gel[29,49,50]. The GIRK chimera, lipids, and MSP1E3 were mixed to respective final concentrations of 10–20 μM, 12 mM, and 100 μM, and incubated at 4 °C for 1.5 h. The GIRK chimera-nanodiscs were assembled by removing the detergent, by adding 80% (w/v) of Bio-Beads SM-2 (Bio-Rad) and mixing at 4 °C for 1.5 h. The GIRK chimera-nanodiscs were purified from aggregates or empty nanodiscs on HIS-Select Nickel Affinity Gel. The His10-tag was cleaved by HRV 3C protease and removed by passage through HIS-Select Nickel Affinity Gel.

**NMR experiments and analyses.** All experiments were performed at 20 °C on Bruker Avance 500 or 600 spectrometers, equipped with a cryogenic probe. All spectra were processed by the Bruker TopSpin 2.1 or 3.1 software, and the data were analyzed using Sparky (T. D. Goddard and D. G. Kneller, Sparky 3, University of California, San Francisco, CA). Protein samples were dissolved in NMR buffer (20 mM HEPES-NaOH, pH 7.0, 150 mM KCl, 0.5 mM GDP, 0.1 mM DSS, and 99.5% D₂O), containing 5 mM DTT (for assignment) or 0.1 mM tris(2-carboxy-yethyl)phosphine (TCEP) (for the other experiment).

Resonance assignments of the Alaβ, Ileδ1, Leuδ, and Valγ methyl groups in Gαi3βγ were obtained by combining mutagenesis and nuclear Overhauser effect (NOE) analyses, based on the crystal structure. To observe the methyl-backbone amide and methyl-methyl NOEs, we acquired a set of three-dimensional NOESY spectra. The [$^1H$–$^1H$] NOESY-[$^1H$–$^{15}N$] TROSY, [$^1H$–$^1H$] NOESY-[$^1H$–$^{13}C$] HMQC, and [$^1H$–$^{13}C$] HMQC-[$^1H$–$^1H$] NOESY-[$^1H$–$^{15}N$] TROSY spectra were recorded on a {uniform(ul)-[$^2H$, $^{15}N$]; Alaβ, Ileδ1-[$^{13}CH_3$]} Gαi3-[non-labeled]βγ sample, with mixing times of 150–200 ms. The [$^1H$–$^1H$] NOESY-[$^1H$–$^{13}C$] HMQC and [$^1H$–$^{13}C$] HMQC-[$^1H$–$^1H$] NOESY-[$^1H$–$^{13}C$] HMQC spectra were recorded on {ul-[$^2H$, $^{15}N$]; Alaβ, Ileδ1, Leu/Val-[$^{13}CH_3$ $^{proS}$-[$^{13}CH_3$]} Gαi3-[non-labeled]βγ and {ul-[$^2H$, $^{15}N$]; Leu/Val-[$^{13}CH_3$, $^{13}CH_3$]} Gαi3-[non-labeled]βγ samples, with a mixing time of 100 ms. The identified NOEs were assigned, based on the crystal structure of Gαi1β₁γ₂ (PDB ID: 1GP2)[33]. For mutagenesis, we constructed 32 mutants of Gαi3 (L5I, A7V, A11V, A12V, V13A, I19V, L23I, A30V, A31V, V50I, V71A, V73I, A98S, A99S, A101V, A111S, A114V, V118F, I162V, V174I, V179I, V185I, V201I, L232I, V233I, L234I, A235V, L273I, L310I, V342I, L348I, and L353I), recorded the $^1H$–$^{13}C$ HMQC spectrum of each mutant in the presence of an excess amount of Gβγ, and compared each spectrum with that of the wild type. We established 96% of the Alaβ (25/25), Ileδ1 (25/26), Leuδ (52/54), and Valγ (39/42) assignments for Gαi3 in complex with Gβγ.

About two-thirds of the resonance assignments of the Alaβ, Ileδ1, Leuδ, and Valγ methyl groups in Gαiqiβγ were readily transferred from those of Gαi3βγ, since the signals from the Gαi3 moiety overlapped. The other signals were assigned by NOE analyses, based on the crystal structure. We recorded the [$^1H$–$^{13}C$] HMQC-[$^1H$–$^1H$] NOESY-[$^1H$–$^{13}C$] HMQC spectra with a mixing time of 100 ms for the {ul-[$^2H$, $^{15}N$]; Alaβ, Ileδ1, Leuδ, Valγ-[$^{13}CH_3$]} Gαiqi-[non-labeled]βγ and {ul-[$^2H$, $^{15}N$]; Leu/Val-[$^{13}CH_3$, $^{13}CH_3$]} Gαiqi-[non-labeled]βγ samples. The $^1H$–$^{13}C$ HMQC spectrum was recorded on a {[ul-$^2H$,$^{15}N$], Ileδ1, Leu/Val$^{proS}$-[$^{13}CH_3$]} Gαiqi-[non-labeled]βγ sample, to obtain the stereospecific assignments for the Leu/Val$^{proS}$ and Leu/Val$^{proR}$ signals. We used the crystal structures of Gαi1β₁γ₂ (PDB ID: 1GP2) and Gαqβ₁γ₂ (PDB ID: 3AH8)[51] as references. We established 95% of the Alaβ (20/23), Ileδ1 (22/22), Leuδ (52/54), and Valγ (40/42) assignments for Gαiqi in complex with Gβγ.

As for Gαi3-q(αA)βγ, the resonance assignments of the signals overlapping with those of Gαi3βγ, were transferred from those of Gαi3βγ.

To examine the chemical shift changes of Gαi3 upon forming the Gαi3βγ complex, we prepared NMR samples containing 100 μM {ul-[$^2H$, $^{15}N$]; Alaβ, Ileδ1, Leuδ, Valγ-[$^{13}CH_3$]} Gαi3 in the GDP-bound form, or 120 μM {ul-[$^2H$, $^{15}N$]; Alaβ, Ileδ1, Leuδ, Valγ-[$^{13}CH_3$]} Gαi3-[non-labeled]βγ (hereafter referred to as Gαi3 [ILVA]βγ), and obtained the $^1H$–$^{13}C$ HMQC spectra for each sample. The chemical shift differences (Δδ) were calculated using the equation $\Delta\delta = [(\Delta\delta_H)^2 + (\Delta\delta_C/5.9)^2]^{0.5}$.

To examine the spectral changes of Gαi3βγ, Gαiqiβγ, and Gαi3-q(αA)βγ induced by the addition of the GIRK chimera-nanodiscs, we prepared NMR samples containing Gα[ILVA]βγ (11 μM), with or without the GIRK chimera-nanodiscs (22 μM), and obtained the $^1H$–$^{13}C$ HMQC spectra for each sample. We also performed experiments using the empty nanodiscs at the concentration that gives a lipid amount similar to that of the GIRK chimera-nanodiscs.

For site-specific spin-labeling, we first prepared the GIRK chimera with the C53S/C310T mutations, as it lacks reactive cysteine residues. Using this mutant as a template, cysteine substitutions were separately introduced to Q344, V351, and L366. MSP1E3, another protein component of the GIRK chimera-nanodisc, has no cysteine residue. After the GIRK chimera-nanodiscs were purified, spin-labeling was performed in buffer, composed of 50 mM Tris-HCl, pH 7.0, 100 mM NaCl, 50 mM KCl, and 0.1 mM TCEP. The GIRK chimera-nanodiscs were incubated with 0.9 mM 4-maleimido-2,2,6,6-tetramethylpiperidine-1-oxyl (4-maleimido-TEMPO) (Sigma-Aldrich) at room temperature for 4 h. Single cysteine labeling was confirmed by MALDI-TOF mass spectrometry on an Axima TOF² mass spectrometer (Shimadzu Biotech). The excess 4-maleimido-TEMPO was removed by passage through NAP-5 or PD-10 desalting columns (GE healthcare).

In the PRE experiments examining the paramagnetic state, $^1H$-$^{13}C$ HMQC spectra were recorded for samples containing 20 μM Gαi3[ILVA]βγ and 25 μM 4-maleimido-TEMPO-labeled GIRK chimera-nanodiscs. Subsequently, the samples were reduced to the diamagnetic state by an incubation at 30 °C for 1 h in the presence of 0.3 mM ascorbic acid, and the $^1H$-$^{13}C$ HMQC spectra were recorded. Using the signal intensities in the paramagnetic state ($I^{para}$) and the diamagnetic state ($I^{dia}$), the PRE contribution to the transverse relaxation rate, $\Gamma_2$, was calculated by the following equation[52]:

$$\frac{I^{para}}{I^{dia}} = \frac{\exp(-\Gamma_2 t_{HMQC}) R_2^{diaH} R_2^{diaHC}}{(R_2^{diaH} + \Gamma_2)(R_2^{diaHC} + \Gamma_2)} \quad (1)$$

where $R_2^{diaH}$ and $R_2^{diaHC}$ are the transverse relaxation rates of the $^1H$ single quantum coherence and the $^1H$-$^{13}C$ multiple quantum coherence of the side chain methyl groups in the diamagnetic state, respectively. The $R_2^{diaH}$ and $R_2^{diaHC}$ rates were measured using NMR samples containing 200 μM Gαi3[ILVA]βγ or 250 μM Gαiqi[ILVA]βγ, with the pulse sequences, which create $^1H$ single quantum or $^1H$-$^{13}C$ multiple quantum coherences during the relaxation periods[53,54]. The magnetization transfer time in HMQC, $t_{HMQC}$, was set to 6.9 ms.

**Structure calculation**. We built a homology model of Gαi3(GDP)β₁γ₂ by SWISS-MODEL[55] using the crystal structure of Gαi1(GDP)β₁γ₂ (PDB ID: 1GP2)[33] as the template. The amino acid sequences are 94% identical between Gαi3 and Gαi1. For the GIRK chimera, the crystal structure (PDB ID: 2QKS_1)[27] does not include the coordinates of residues 75–81 and 365–371 of GIRK1. We thus modeled residues 75–81 as a loop by SWISS–MODEL[55], and transferred the coordinates of residues 365–370 from the crystal structure of the GIRK1 cytoplasmic region (PDB ID: 1N9P)[56] by superposition. The consequent structure was embedded in a nanodisc using CHARMM-GUI Nanodisc Builder[57]. These structures were used for structural calculations with XPLOR-NIH[58,59] and figure preparation.

The Gln344, Val351, and Leu366 residues of each subunit of the GIRK chimera were replaced with cysteine residues conjugated to 4-maleimido-TEMPO. A five-conformer ensemble of the 4-maleimido-TEMPO group was modeled by a simulated annealing procedure, in order to represent the conformational space sampled by the flexible paramagnetic probes. The construction of the complex model structure was conducted in a two-step manner. In the first step, we docked the C-terminal helix of GIRK to Gαi3, using the PRE data from L366C-TEMPO. To reduce the computational load, only the C-terminal helix of GIRK from one subunit and Gαi3 were subjected to the docking simulation. Rigid body docking of the two segments was performed by three successive simulated annealing stages, in order to avoid the direct contacts of the 4-maleimido-TEMPO groups with Gαi3. In the first stage, the two segments were docked using a target function containing a PRE potential term and a van der Waals repulsion term, by 10 ps of torsion angle dynamics at 3000 K followed by slow-cooling to 10 K. In the second stage, the two segments were allowed to move under a target function containing the van der Waals repulsion term and a pseudo NOE potential term that pushes the side chain of L366C-TEMPO apart from the interface, by 3 ps of torsion angle dynamics at 3000 K followed by slow-cooling to 10 K. In the final stage, the segments were docked again with a target function containing the PRE potential term, the van der Waals repulsion term, and an implicit solvent potential term[60], by 1 ps of torsion angle dynamics at 3000 K followed by slow-cooling to 10 K and further minimization at 10 K. In all the three stages, the usual terms to retain the plausible bond lengths, angles, planarity, and torsion angles of the polypeptide were appended to the target functions. Five hundred complex structures were generated, and the lowest energy structure was selected to be restored into the full-length GIRK chimera-nanodisc and Gαi3βγ by superposition. In the following calculations, Gαi3βγ was treated as a stable complex and the position, so the orientation and position of Gβγ could be simultaneously fixed on the basis of the distance information observed for Gαi3. In the second step of the calculation, the flexible region connecting the C-terminal helix and the β-strand region of the GIRK chimera (residues 352–357) was randomized. It is reasonable to expect that this region is flexible, because residues 352 to 357 adopt a random coil conformation and form few hydrogen bonds with other structural elements, as observed in the crystal structures[28,56] and molecular dynamics simulations[61] of GIRK. These residues were randomized in the torsion angle space while the relative coordinates of the C-terminal helix against Gαi3βγ were fixed, and the structures not exhibiting clashes between molecules were stored. A total of 30,000 structures were generated, and the Γ₂ values from Q344C-TEMPO, V351C-TEMPO, and L366C-TEMPO of all four subunits were back-calculated for each structure. Ensembles were built using all or part of these structures, and the relative populations of the structures were then optimized to minimize the ensemble averaged Q-factor, $Q_{ens}$, defined by the following equation:[38]

$$Q_{ens} = \sqrt{\frac{\sum_i \left( \Gamma_2^{obs}(i) - \sum_k p_k \Gamma_2^{calc,k}(i) \right)^2}{\sum_i \Gamma_2^{obs}(i)^2}} \quad (2)$$

, where $\Gamma_2^{obs}(i)$ is the experimental $\Gamma_2$ of the $i$th residue, $\Gamma_2^{calc,k}(i)$ is the $\Gamma_2$ of the $i$th residue back-calculated from the $k$th structure, and $p_k$ is the population of the $k$th structure. The minimization was performed by using the optim function implemented in the R language (https://www.r-project.org/), in a stepwise manner. In a typical minimization procedure using 10,000 structures, the populations of 100 randomly chosen structures were optimized by the successive applications of the simulated annealing method and the Broyden–Fletcher–Goldfarb–Shanno method. Subsequently, 10 groups of these 100 structures were merged and the same optimization was performed against 1000 structures, by using 1/10 of the population of each structure obtained in the previous calculations as the initial values. The ten top percent of the structures with the highest populations were selected from each group containing 1000 structures to obtain a total of 1000 structures. The final optimization was performed by using 1/10 of the population of each structure obtained in the previous calculations as the initial values. We confirmed that 10,000 structures were sufficient to represent the possible spatial distribution of Gαi3βγ, because the overall Q-factor did not change much when we used all 30,000 structures to build an ensemble (Supplementary Fig. 7).

To visualize the distribution of Gαi3βγ relative to GIRK, 10 ensembles, each consisting of 1000 structures, were generated and subjected to the reweighted atomic probability density map calculation using XPLOR-NIH[39]. The probability maps were rendered using PyMOL (http://www.pymol.org). The electrostatic potential surfaces of the molecules were generated using the APBS Tools 2.1 plugin for PyMOL[62].

**Reporting summary**. Further information on research design is available in the Nature Research Reporting Summary linked to this article.

## Data availability

Sequence information on human GIRK1, GIRK2, GRK3, DOR, D2R, Gαi3, Gαq, Gβ₁, Gγ₂, PTX S1, mouse GIRK1, and KirBac1.3 are available in the UniProt Knowledgebase under accession codes P48549, P48051, P35626, P41143, P14416, P08754, P50148, P62873, P59768, D2WF63, P63250, and Q146M9. The PDB accession codes 1GP2, 3AH8, 2QKS, 1N9P, 1GIA, 4KFM, and 1AZT were used in this study. The source data underlying Fig. 1–4, Supplementary Figs. 2, 3, 5, 7, and 10 are provided as a Source Data file. All other data are available from the corresponding author upon reasonable request.

## Code availability

Codes for structural calculations are available from the corresponding author upon request.

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

## Acknowledgements

We thank Mr. Qingci Zhao and Dr. Noritaka Nishida for helping with the culture of HEK293T cells for the BRET assays. The BRET assays were performed at the One-stop Sharing Facility Center for Future Drug Discoveries in the Graduate School of Pharmaceutical Sciences, The University of Tokyo. This work was supported in part by grants from the Japan New Energy and Industrial Technology Development Organization and the Ministry of Economy, Trade, and Industry (to I.S.); Japan Agency for Medical Research and Development (AMED) Grant Number JP18ae010104 (to I.S.); Japan Society for the Promotion of Science (JSPS) KAKENHI Grant Numbers JP17H03978 (to M.O.), JP18H04679 (to M.O.), JP17H06097 (to I.S.), and JP18J13147 (to H.K.); a grant from The Vehicle Racing Commemorative Foundation (to M.O.); and a grant from Takeda Science Foundation (to M.O.).

## Author contributions

H.K., Y.T., Y.M., M.O., and I.S. designed the study; H.K. purified proteins, constructed the mutants, and performed BRET assays with advice from Y.T.; H.K. and Y.T. performed the NMR experiments; H.K. and S.I. performed structural calculations; Y.I., Y.M., and M.Y. constructed expression vectors and performed preliminary NMR experiments at the early stage of the study; and H.K., Y.T., S.I., M.O., and I.S. analyzed the data and wrote the manuscript.

## Additional information

**Competing interests:** The authors declare no competing interests.

