## [Peer Review File · Nature Communications]

Reviewers' comments:

Reviewer #1 (Remarks to the Author):

In their paper entitled "Structural mechanism underlying G protein family-specific regulation of G protein-gated inwardly rectifying potassium channel", Kano et al describe a possible interaction between the alpha-helical domain of an inhibitory G-alpha subunit and the C-terminus of the GIRK ion channel. The authors claim that this interaction is the key determinant for the selectivity of GIRK inhibition by G-beta-gamma subunits originating from GPCR-activated Galpha,i/o-beta-gamma complexes and not by G-alpha,q or s subunits.

The authors use in vivo BRET experiments in HEK293T cells to determine the complex formation between G-beta-gamma and the GIRK channel or GRK upon activation of an opioid receptor. By normalizing the BRET effects with GRK, the authors come up with a specificity factor for the desired signaling pathway. By using different G-alpha subunits and chimeras, they showed that GIRK inhibition is more specific if a G-alpha,i subunit is present in the heterotrimer, compared to G-alpha,q. These functional studies are followed up by NMR titration experiments with a heterotrimeric G-protein and a GIRK chimera in nanodiscs and finally the authors come up with a structural model of a putatively functional complex between GIRK and G-protein and possibly also with a GPCR. This model is used to explain the selectivity of GIRK inhibition by inhibitory G-alpha subunits.

This paper contains a large set of both, biological as well as NMR structural data. The introduction and the discussion parts are a bit too long. Except for a few typos, the paper is well written and otherwise concise.

Overall, it remains a bit puzzling, how the very weak interaction between G-alpha and GIRK can determine the specificity for the much more tightly interacting G-beta-gamma subunits to the ion channel. The in vivo and NMR results may suggest a specific interaction, however, the effects are generally very small and the NMR intensity changes are quite minor in all cases. In this respect a titration at lower ionic strengths of the buffer would confirm the charge-mediated interaction and would probably lead to a more specific interaction pattern. Furthermore, the most pronounced intensity reduction ratios in Fig. 2, do not cluster around the proposed site on the surface of the helical domain but rather to the Ras domain in close proximity to the nucleotide binding site. So, how was the binding site in the helical domain identified? In addition, a control experiment with empty nanodiscs would be suitable as a reference to clearly separate NMR effects induced by the membrane alone from those induced specifically by GIRK. If G-alpha is the driving force for the initial interaction of the heterotrimer with GIRK, the same NMR spectral changes should be visible with isotope-labeled G-alpha alone upon addition of GIRK nanodiscs.

The NMR methyl assignment work is impressive and could be compared to a previous study on G-alpha,i1. Having assigned G-alpha methyls in the free and G-beta-gamma-bound forms (Figs. S3 & S4), it would be very informative to compare these spectra to probe changes in the conformation of G-alpha in the heterotrimer.

Another open question is whether the three constructs (in particular the chimeras) used to probe the specificity in the BRET assay show comparable nucleotide binding properties and GPCR coupling efficiency. Recent cryo-EM structures showed that not only the C-terminal 4 amino acid residues are required for GPCR interaction. Thus, the G-alpha-qi5 construct might not be activated by the chosen GPCRs. Also, the G-alpha-iqi might show altered nucleotide binding properties and thus altered dissociation properties of the G-beta-gamma subunit.

The obtained structural model is based on PRE experiments with three single cysteine variants shows that the main interaction is mediated by the C-terminal tail of GIRK. The large dynamics of the complex is accounted for by using structure ensembles for the back-calculation of the PRE effects. This yields a structural ensemble exhibiting still very large changes in the orientation of the individual members (Fig. S6), limiting the value of such a sophisticated calculation. In the end, the question is how relevant the three poses shown in Fig S7 are for the mechanism of GIRK activation. Furthermore, how was the relative orientation of the N-terminal G-alpha helix and G-beta defined? These features cannot be obtained with the described PRE experiments where only

G-alpha is labeled with isotopes.

The paper would also benefit from a more thorough discussion on other complexes formed by G-alpha and G-beta-gamma upon activation by a GPCR. How likely is an association with GIRK considering the presence of partner proteins like adenylate cyclase and others? How could this be integrated into the model shown in Fig. 5 ?

Reviewer #2 (Remarks to the Author):

Kano et al, 'Structural mechanism underlying G protein family-specific regulation of GIRK

Over the years the group of Shimada and co-workers has spent considerable efforts in elucidating the mechanism of action of GIRKs. Following several earlier publications this manuscript is a continuation of this solid work by the group. In the current manuscript the authors are using a combination of BRET and NMR spectroscopy to investigate the specific regulation of GIRK via pathways that are triggered through the neurotransmitter channels related to inhibitory GPCR signalling. The intriguing question the authors are investigating is why the activation of GIRK occurs only via G $\beta\gamma$ subunits that are related to the inhibitory pathway i.e. via the Gai/o domain of the heterotrimeric G protein but not with other heterotrimeric G protein Ga domains e.g. from Gs or Gq. Over recent years this observation has resulted in the proposition of many model mechanisms, however, based on very little experimental evidence. For the first time now Kano et al. are providing experimental evidence, which shows that the preassembly of a complex between the heterotrimeric G protein containing the Gai/o subunit with GIRK is key and responsible for the specific regulation of the potassium channel via the G $\beta\gamma$ subunit within the same heterotrimer. A comparable interaction of non-Gai/o containing heterotrimeric G proteins simply shows insufficient affinity for relevant association. Hence, a preassembly via Gai/o - GIRK interaction provides the activating G $\beta\gamma$ subunits via a close proximity mechanism. This is important for the fast activation of GIRK and explains why this can be achieved rapidly.

In view of the relevance of GIRK in the involvement of multiple disorders e.g. epilepsy etc, mechanistic studies can contribute towards the development of drug based therapies making this contribution an important study which provides strong evidence towards the understanding of how GIRK gets specifically activated. This is new insight into the activation of this potassium channel group and settles many years of speculation. The proposed regulatory mechanism obtained from the in vitro study would explain the efficient and rapid control of GIRK in a physiological environment.

While initial tests by the authors use BRET to establish independently the involvement of the Gai/o subunit of the heterotrimeric G protein in the interaction with GIRK, the major investigation is concentrating on NMR spectroscopy, which provides structural evidence for complex formation at a resolution which allows to propose which parts of the Ga domain are interacting with the cytoplasmic GIRK. In view of the dynamic nature of the problem this is a very appropriate choice of technique, however, the authors have to overcome some steep hurdles in view of the 300 kDa size of the protein complex assembly in nanodiscs. The authors are conducting this interaction with GIRK embedded in a lipid raft to emulate the properties of the membrane bilayer. In this setting the Ga-GIRK interaction proves to be weak so that it is observed in fast exchange with the G protein uncoupled from GIRK. Therefore this leads to only very small chemical shift changes and Kano et al therefore use intensity variations to map the interaction. Questions remain as to the contribution of the N-terminal involvement of Ga with the lipid bilayer and how this contributes to binding but might also interfere with the simple data analysis conducted based on small chemical shift changes and intensity variations. However, the additional use of experimental distance restraints obtained from PRE measurements enabled the authors to build a model of the complex and to propose an overall mechanism of action. This work is complex in its investigations and

relies on multiple isotopically labelled proteins and a careful choice of investigative strategy that shows that NMR is a key player in the investigation of protein interactions in the vicinity of the membrane, in particular where intrinsic protein dynamics are relevant to form and modulate interactions. In contrast to techniques such as crystallography or cryo-EM that put a strong bias on their outcomes there is no such bias by NMR and it is the method of choice.

This is beautiful work, very well executed and presented and I would like the authors to address a range of criticisms prior to publication:

1) Nanodiscs provide a lipid environment, which can also incorporate hydrophobic entities such as e.g. spin labels. This raises concerns with regard to the interpretation of PRE measurements. Can the authors convincingly show that there was no spin label contamination into the nanodiscs and that the measured contributions to PRE are the specific result of TEMPO attached to GIRK?

2) The authors are using a BRET assay to determine the specificity of different Ga domains in activating GIRK. The outcome of two experiments is compared: one which provides the Ga specific GIRK binding to Gβγ while the other one gives the GPCR related signaling level due to βγ interaction with a GRK. These measurements are reflecting the thermodynamic stability of the interactions and accordingly thermodynamic control. However, as fast switching of the GIRK seems to be a feature of the mechanism it could be that the channel activation through Gβγ is under kinetic control and hence not related to the BRET values measured. The authors should comment on this possibility.

3) The NMR investigations are conducted with GIRK embedded in a nanodisc environment. This allows to approximate various effects that might be related to protein encounters in the membrane proximity and to mimic influence related to the Ga N-terminal interaction with the bilayer. This all makes sense and is a good choice in order to project the results of the in vitro investigations into a physiological environment. However, typically NMR work uses rather small nanodiscs to reduce detrimental effects related to fast spin relaxation. I simply wonder about the geometric constraints here. It would be useful if the authors could provide some information on the physical size of the nanodisc diameter and the size of the interacting proteins, so that one can conclude on the physical size requirements of the complex and assess whether any interference from the nanodisc scaffold protein might have influenced the study.

4) Figure 2C shows the reduction in intensity rather than a simple intensity ratio, which is typically shown in such situations. It might be less confusing to show the latter and the figure and text should be adapted accordingly.

5) On several occasions there is a relatively diffuse mention of Ga_{i/o} N-terminal interaction with the lipid bilayer, which to some extent mimics the anchoring of the heterotrimeric G protein to the membrane. The authors could not closely simulate the full anchoring taking place in the cellular environment as they found that the corresponding lipidated G protein heterotrimer aggregated. Hence, left only with the N-terminus supposedly interacting with the bilayer one has to assume that there is sufficient affinity for Ga to 'stick' to the nanodisc lipid bilayer. There is no quantification provided. Based on that, it is not immediately obvious to me why the intensity comparison was conducted with reference to the freely soluble Gaβγ domain, in absence of nanodiscs. I would have thought that comparing it to the intensities in the presence of nanodiscs would provide a better reference for signal intensities. Using the latter would improve the confidence in the observed intensity changes upon GIRK binding. Likewise, it would also allow a better estimate of the K_d, currently put at 200 μM.

6) The chemical shift changes shown in Fig 2 are with < 0.01 ppm very small. Based on these small changes, how is it possible to get further intensity reductions due to differential line broadening, as the authors claim. On p.22-23 it is argued that some of the signal intensity drops observed for interacting residues are related to differential line broadening, postulating that exchange is on the slow to intermediate side of coalescence. On what experimental evidence is the latter assumption residing? The chemical shift changes shown in Fig. 2 are very small (quote < 0.01 ppm), hence, very likely the exchange should be on the fast exchange side of signal coalescence. I notice that two spectrometer field strengths were used in the study. Has field

dependence been noticed validating the author's assumptions? An alternative explanation for the increased intensity drops could be related to anisotropic tumbling accentuated in the bound state. Such anisotropic tumbling should already be visible in the nanodisc-associated state of the heterotrimeric G protein as facilitated via the N-terminal interaction of the G α i with the lipid bilayer. This seems to be another good reason why using the latter state as a reference for the intensity comparison should be a better choice (see point 5).

Reviewer #3 (Remarks to the Author):

G $\beta\gamma$ subunits released selectively from G α i/o-type G-proteins activate GIRK channels, which led to the proposal that GPCRs, G protein and GIRK channels are compartmentalized or preassembled. In this manuscript, the authors performed BRET assays in transfected cells and NMR experiments with reconstituted GIRK channels in nanodiscs to study which part of the heterotrimeric G protein determines assembly with GIRK channels. They propose that the negatively charged C-terminal helix of GIRK channels forms a binding interface with the positively charged α A helix in the helical domain of G α i. They conclude that this interface is responsible for G-protein/GIRK complex formation and the G-protein selectivity of GIRK activation. The paper provides new information on the G α i/GIRK interface, which however needs further validation. In addition, the authors provide a model that can explain how the activated G protein can gate GIRK channels while being tethered to the channel.

Major

The authors use a BRET assay monitoring G $\beta\gamma$ /GIRK interaction as a surrogate for directly measuring K⁺ currents. This allows them to normalize GIRK activation to G protein activation, which is measured by G $\beta\gamma$ /GRK BRET. This normalization is appealing but may also be problematic. For example, it is unclear whether G $\beta\gamma$ binding to GRKs does not exhibit some G α subunit preference, similar as the GIRKs do. Moreover, since according to the authors already the heterotrimeric G protein assembles with GIRKs (the schemes in Figure 1A, C are in that sense misleading, compare with Figure 5a) it is important to know how much basal G $\beta\gamma$ /GIRK BRET is obtained with the different G proteins used. One would expect a higher basal BRET for G $\beta\gamma$ /GIRK with G proteins that bind GIRKs. Therefore, the basal BRET before agonist stimulation should be provided for all BRET experiments. Likewise, it is unclear whether the Δ BRET between G $\beta\gamma$ /GIRK monitors the extent of GIRK activation. A small Δ BRET upon agonist application may, e.g. also reflect high tonic GIRK activity (high basal BRET) that cannot be stimulated by agonist to the same extent as in the absence of tonic activity (Naloxone is a neutral antagonist and therefore does not reveal tonic activity in the system). Given these caveats of the BRET assay, I believe that a direct electrophysiological read-out for GIRK activity should be used to, at least, validate the main conclusions of the manuscript (see below).

The most important finding is that the α A helix in the helical domain of G α i forms the binding interface with GIRKs. The only validation that is provided is a loss-of-function experiment showing a decline in the specificity factor for G α i3-q(α A), which was significantly smaller than that of the wild-type G α i3. It would be more convincing to use a gain-of-function experiment to support the main conclusion of the manuscript and to show that G α q (or G α s) with the α A helix of G α i shows an increase in the specificity factor. Moreover, gain- and loss-of-function mutants should also be analyzed using NMR and GIRK nanodiscs. This will reveal whether the mutants exhibit the expected shifts in intensities. Given that the BRET experiments only provide an indirect measure for GIRK activation, it is crucial to directly measure whether gain- and loss-of-function mutants activate K⁺ currents in a heterologous expression system. This will give the direct information on whether the mutants activate GIRKs and/or influence current activation kinetics in the expected manner.

I am surprised that no larger spectral changes were observed with G α iqi[ILVA] $\beta\gamma$ in the N-terminal region, which I would expect to interact with membrane lipids of the nanodiscs even though G α iqi[ILVA] $\beta\gamma$ does not bind GIRKs (Fig. 2C).

Minor

The abstract needs rewriting, as it is too generic and does not clearly highlight the novel insights

this study provides. Several labs have shown that Gα_i binds GIRK subunits and that this interaction contributes to the specificity of GIRK activation. Likewise, the Gα helical domain was implicated in this interaction before. This is not really novel and should therefore not be the main conclusion of the abstract. In my opinion, the most significant finding is the identification of a binding interface, which should be clearly stated in the abstract (and accordingly thoroughly validated, see above).

P3 "...thus regulating heart rate and inhibitory neurotransmission". GIRKs regulate excitatory neurotransmission as well, e.g. by shunting of EPSCs

Note added in the revision process

Touhara and MacKinnon reported a paper regarding the $G\alpha$ family specificity, just after our initial submission (Touhara and MacKinnon, eLife 2018). In this paper, they demonstrated that the rate of $G\beta\gamma$ release is higher in M2R (i/o-coupled) than that in β 2AR (Gs-coupled), and concluded that this kinetic difference accounts for the $G\alpha$ family specificity.

Although they successfully modeled the $G\alpha$ family-specific activation of GIRK on the basis of this notion, they pointed out that GPCRs and G proteins need to be more highly co-localized than expected to form a “hot-spot” region, in order to quantitatively explain the experimentally observed GIRK currents. This co-localization is well-explained in our model, where G proteins, GIRK, and probably GPCRs gather to form a signaling complex, and the local concentrations of these molecules are maintained at high levels. Therefore, we believe our results do not conflict with their model.

Regarding this point, we added the following note in the discussion section.

(Page 20 in the supplementary information)

Several lines of evidence have indicated that the signaling complex also contains GPCR, along with G protein and GIRK¹⁵⁻¹⁷. The GPCR-binding site on $G\alpha\beta\gamma$ is the α 5 helix in the GTPase domain of $G\alpha$ ⁶⁷, and since this region is not involved in the interaction with GIRK, GPCR can also bind to $Gai/o\beta\gamma$ in the proposed complex structure (Supplementary Figure 12). These observations suggest that the $Gai/o\beta\gamma$ -GIRK complex revealed here represents part of an even larger complex including GPCR. **By forming this complex, the local concentrations of GPCR, GIRK, and G proteins are maintained at high levels, which would facilitate the efficient activation of GIRK by increasing the local concentration of $G\beta\gamma$ upon the activation of GPCR. This notion is consistent with the model recently proposed by Touhara and MacKinnon, in which GPCR and G proteins must be highly co-localized in a hot-spot region, in order to achieve sufficiently high concentrations of $G\beta\gamma$ to activate GIRK¹⁴.**

Point-by-point responses to reviewers

We would like to thank all of the reviewers for their thoughtful comments and suggestions on our manuscript. According to the reviewers' comments, we revised the manuscript carefully. Our point-by-point responses are summarized below. The reviewers' comments are shown in *Italic font*, and the textual changes in the revised manuscript are highlighted in red font. We hope that the revised manuscript is considered acceptable for publication.

Reviewer 1

This paper contains a large set of both, biological as well as NMR structural data. The introduction and the discussion parts are a bit too long. Except for a few typos, the paper is well written and otherwise concise.

We appreciate the reviewer's comment. We shortened the introduction section, and moved some topics in the discussion section to the supplementary discussion.

1-1) Overall, it remains a bit puzzling, how the very weak interaction between G-alpha and GIRK can determine the specificity for the much more tightly interacting G-beta-gamma subunits to the ion channel. The in vivo and NMR results may suggest a specific interaction, however, the effects are generally very small and the NMR intensity changes are quite minor in all cases.

We appreciate the reviewer's comment. As the reviewer pointed out, the interaction between G α 3 β γ and GIRK is very weak, with an apparent K_d larger than 200 μ M in our NMR experiments. This apparently low affinity is attributed to the fact that G α 3 β γ is not permanently attached to the lipid membrane in our NMR experiments. Under these conditions, G α 3 β γ and GIRK-nanodiscs freely translocate and tumble, and are less likely to encounter each other in the same orientation as on the membrane, thus leading to the substantial decrease in the apparent binding affinity. It should be noted that the interaction is weak but quite specific, because we did not observe the NMR spectral changes when we used the G α mutant proteins, such as G α 1q and G α 3-q(α A), that do not provoke the activation of GIRK (please also refer to our response to 3-5, raised by Reviewer 3).

We estimate that the actual K_d of the G α 3 β γ -GIRK interaction in the *in-situ* membrane environment is around 200 nM, which is about 1,000-fold stronger than that observed in the NMR experiment. This estimation is based on the previously reported theoretical framework and our previous result that the solubilized G β γ exhibited 10^3 - to 10^4 -fold weaker binding affinity for GIRK than in the native membrane environment (Grasberger et al., 1986; Yokogawa et al., 2011). We expect that this estimated K_d value is sufficiently strong for G α β γ to be tethered to GIRK, and confers the specificity for GIRK activation.

1-2) In this respect a titration at lower ionic strengths of the buffer would confirm the charge-mediated

interaction and would probably lead to a more specific interaction pattern.

We appreciate the reviewer's suggestion. An NMR experiment under low ionic strength conditions is an effective approach to characterize an inherently weak charge-mediated interaction. However, in our case, the positively charged N-terminal helix of G α , which is considered to form a charge-mediated interaction with the negatively-charged membrane lipid (POPG and PIP₂), is highly susceptible to non-specific interactions with the negatively-charged patches on the protein surfaces. Therefore, to faithfully characterize the specific GIRK-G $\alpha\beta\gamma$ interaction and the G $\alpha\beta\gamma$ -membrane interaction, we have to use the GIRK in the membrane environment, and to conduct the binding experiments in the buffer with the physiologically-relevant salt concentration.

Regarding this point, we added the explanation for the buffer conditions as follows.

(Page 10, lines 201-207)

We analyzed the interaction by using solution NMR techniques, which can characterize weak protein-protein interactions in physiological solution environments. In the analyses, we used a recombinant G $\alpha i3\beta\gamma$ that lacks the lipid modification, which is partially localized to the lipid bilayer surface of the nanodiscs via an N-terminal polybasic region^{30,31}. **The experiments were conducted under physiologically-relevant ionic conditions (KCl =150 mM), to suppress the non-specific binding mediated by these polybasic, positively charged regions.**

1-3) Furthermore, the most pronounced intensity reduction ratios in Fig. 2, do not cluster around the proposed site on the surface of the helical domain but rather to the Ras domain in close proximity to the nucleotide binding site. So, how was the binding site in the helical domain identified?

We appreciate the reviewer's comment. We identified the binding site on the basis of the site-specific PREs observed on G α . The methyl groups with significant Γ_2 values from L366C-TEMPO were clustered around the αA and αB helices in the helical domain of G $\alpha i3$, while a few methyl groups exhibited Γ_2 values larger than 5 s⁻¹ from Q344C-TEMPO and V351C-TEMPO. On the basis of these PRE patterns, we concluded that L366 of GIRK is proximate to the helical domain of G $\alpha i3$ in the complex, while the other regions of GIRK do not form stable interactions with G $\alpha i3\beta\gamma$.

The methyl groups with pronounced intensity reduction ratios were not clustered around the helical domain. We have the following explanations for this result.

- (i) The methyl groups are sparsely distributed around the proposed GIRK-binding site on the helical domain. The methyl groups on the αA helix (V71 γ 1,2, V72 γ 1,2, V73 γ 1,2, I78 δ 1, I81 δ 1, I82 δ 1, A83 β , I84 δ 1, I85 δ 1, A87 β) do not face toward the GIRK binding surface.
- (ii) The site-specific intensity reductions are mainly caused by the ¹H exchange broadening effects, because these effects result in signal broadenings in both the F1 and F2 directions (please also refer

to our reply to 2-6-1). These ^1H exchange broadening effects on the methyl groups are mainly dominated by the differences in the ring-current effects from the nearby aromatic amino acids. In the proposed GIRK-binding site on the helical domain, there are few aromatic amino acids that can cause these ring current shifts (F108 and Y146, neither of which directly contacts with GIRK). In contrast, aromatic amino acids are clustered around the $\text{G}\beta\gamma$ -binding site (F199, F215, W211, and W258 on $\text{G}\alpha$; W99 and Y145 on $\text{G}\beta\gamma$) and the N-terminal region (W82 and Y85 on $\text{G}\beta\gamma$), where relatively large intensity reductions were observed. For this reason, the ^1H exchange broadening effects were relatively small around the GIRK-binding site on the helical domain, while these effects were significantly observed in the N-terminal region and the $\text{G}\beta\gamma$ -binding site.

Supporting Figure 1 (A) Distributions of the observed methyl groups and aromatic amino acids around the binding site for GIRK in the structural model of the complex. (B) Distributions of the aromatic amino acids around the N-terminal helix and the $\text{G}\beta\gamma$ -binding site, where relatively large intensity reductions were observed upon the addition of the GIRK-chimera nanodiscs.

1-4) In addition, a control experiment with empty nanodiscs would be suitable as a reference to clearly separate NMR effects induced by the membrane alone from those induced specifically by GIRK.

We appreciate the reviewer's suggestion. According to the reviewer's suggestion, we conducted a control experiment using empty nanodiscs. Upon the addition of the empty nanodiscs, we did not observe significant intensity reductions of $\text{G}\alpha\text{i}3[\text{ILVA}]\beta\gamma$. This result indicates that the interaction with the membrane lipid alone is not strong enough to facilitate the formation of the complex, and that the population of $\text{G}\alpha\text{i}3[\text{ILVA}]\beta\gamma$ bound to the empty nanodiscs is so small that we could not detect the NMR spectral changes. We should note that this result does not indicate the absence of the

interaction with the membrane lipid, because we clearly observed significant intensity reductions in the NMR signals from the N-terminal lipid-binding surfaces of Gai3[ILVA] $\beta\gamma$ upon the addition of GIRK-nanodiscs, and we also estimated that the affinity of Gai3 $\beta\gamma$ for GIRK is enhanced by about 2-fold by the interaction between Gai3 $\beta\gamma$ and the nanodisc membrane, on the basis of the intermolecular PRE experiments using GIRK solubilized in detergent micelles (please also refer to our response to 2-5-1, raised by reviewer 2).

Although we were not able to clearly separate the effects induced by the membrane alone and those induced specifically by GIRK, these results clearly support our notions that Gai3 $\beta\gamma$ must interact with both GIRK and the nanodisc membrane to stably form the Gai3 $\beta\gamma$ -GIRK complex, and that the observed intensity reduction of Gai3[ILVA] $\beta\gamma$ reflects the specific interaction between Gai3 $\beta\gamma$ and GIRK-nanodiscs.

In the revised manuscript, we added the results of the control experiment with the empty nanodiscs.

(Page 11, lines 220-234)

Upon the addition of 2 equivalents of the GIRK chimera-nanodiscs to Gai3[ILVA] $\beta\gamma$, most of the signals exhibited intensity reductions with relative intensities lower than 0.9, and the signals from L5 δ 1, L5 δ 2, A12 β , V13 γ 1, V13 γ 2, A30 β , A31 β (N-terminal helix), L36 δ 1, L36 δ 2, L37 δ 1, L37 δ 2 (β 1 strand), A41 β (β 1- α 1 loop), I127 δ 1 (α C helix), L148 δ 1 (α D- α E loop), L159 δ 2 (α E helix), V218 γ 2 (α 2- β 4 loop), I221 δ 1 (β 4 strand), L232 δ 2, L234 δ 2 (β 4- α 3 loop), L249 δ 1, L249 δ 2, I253 δ 1 (α 3 helix), I264 δ 1, I265 δ 1 (β 5 strand), I278 δ 1, L283 δ 1 (α G- α 4 loop), and L348 δ 1 (C-terminus) exhibited further reduced intensities lower than 0.8 (Figures 2A-C), while the observed chemical shift changes were very small (< 0.01 ppm). **When we added the empty nanodiscs, we did not observe significant intensity reductions, showing that the observed intensity reductions upon the addition of the GIRK chimera-nanodiscs are mainly triggered by the specific binding of Gai3[ILVA] $\beta\gamma$ to the GIRK chimera (Figure 2C).** The overall intensity reductions are caused by slower tumbling, due to an increased average molecular weight, indicating that a fraction of Gai3[ILVA] $\beta\gamma$ forms a complex with the GIRK chimera-nanodiscs.

Figure 2 NMR spectral changes of $G\alpha 3[ILVA]\beta\gamma$ induced by the GIRK chimera-nanodiscs. (A) Overlay of the 1H - ^{13}C HMQC spectra of $G\alpha 3[ILVA]\beta\gamma$ in the presence (red) and absence (black) of 2 eq. of the GIRK chimera-nanodiscs. (B) Close-up views and cross-sections of V13 γ 2, L148 δ 1, and L310 δ 2 as representative signals. For comparison, the corresponding signals of $G\alpha qi[ILVA]\beta\gamma$ in the presence (blue) and absence (black) of the GIRK chimera-nanodiscs are shown on the right. (C) Plots of relative intensities of $G\alpha 3[ILVA]\beta\gamma$ (top) and $G\alpha qi[ILVA]\beta\gamma$ (middle) in the presence of 2 eq. of the GIRK chimera-nanodiscs. A plot of the relative intensities of $G\alpha 3[ILVA]\beta\gamma$ in the presence of empty nanodiscs is also shown (bottom). The error bars are calculated based on the signal-to-noise ratios. Methyl groups with relative intensities lower than 0.80 are colored according to their values. (D) Mapping of the methyl groups of $G\alpha 1\beta\gamma$ with significant intensity reductions on the structure of $G\alpha 1\beta\gamma$ (PDB ID: 1GP2)³³. Methyl groups are shown as spheres and colored according to their relative intensity values. Source data are provided as a Source Data File.

1-5) If G -alpha is the driving force for the initial interaction of the heterotrimer with GIRK, the same NMR spectral changes should be visible with isotope-labeled G -alpha alone upon addition of GIRK nanodiscs.

We appreciate the reviewer's comment. As the reviewer pointed out, the interaction between GIRK and $G\alpha$ alone in the GTP-bound form ($G\alpha(GTP)$) actually occurs, which we characterized on the basis of the intermolecular PREs and transferred cross saturation experiments in our previous study (Mase et al., 2012). In this study, significant intermolecular PREs were observed in the C-terminal

region of GIRK from the spin label introduced to Cys82 of $G\alpha$ (GTP), which is located on the helical domain. The proximity of these two regions is similarly observed in our NMR experiments observing the GIRK- $G\alpha\beta\gamma$ interaction, supporting our hypothesis that the $G\alpha$ is the driving force for the initial interaction with GIRK. We note that the overall binding poses are very different between the $G\alpha$ -GIRK and $G\alpha\beta\gamma$ -GIRK complexes, because the GTPase domain, which is covered by $G\beta\gamma$ in the $G\alpha\beta\gamma$ form, can also bind to the C-terminal region of GIRK in $G\alpha$ (GTP).

Supporting Figure 2 The interactions formed between the helical domain and the C-terminal region of GIRK. (Left) The interacting sites on $G\alpha\beta\gamma$ and GIRK (green), identified in the current study. (Right) The interacting sites on $G\alpha$ (GTP) and GIRK (red), identified in our previous study (Mase et al., 2012). The $G\beta\gamma$ binding site on GIRK is also shown in blue, and was identified in our previous study and the crystal structure of the $G\beta\gamma$ -GIRK complex (Whorton and MacKinnon, 2013; Yokogawa et al., 2011).

Regarding the interaction between $G\alpha$ alone and GIRK, we added the following sentences to the discussion.

(Page 18, lines 384-394)

Closer inspection of the complex structure also revealed the charge complementarity between the positively charged αA helix of $G\alpha i3$ and the negatively charged C-terminal helix of GIRK (Supplementary Figure 11). Since the charge distributions around the αA helix significantly differ among the $G\alpha$ families, this charge complementarity is likely to promote the formation of the $G\alpha i/o\beta\gamma$ -GIRK complex in a family-specific manner. We note that the interaction between the αA helix of $G\alpha i3$ and the C-terminal region of GIRK was similarly observed in a $G\alpha i/o$ (GTP)-GIRK complex formed during the activation of GIRK, which we previously characterized by intermolecular PRE and transferred cross-saturation experiments²¹. This observation indicates that the helical domain of $G\alpha i3$, where the αA helix is located, has an inherent affinity for GIRK, and that the interaction is the driving force for the interaction between $G\alpha i3\beta\gamma$ and GIRK.

1-6) The NMR methyl assignment work is impressive and could be compared to a previous study on G-alpha,i1. Having assigned G-alpha methyls in the free and G-beta-gamma-bound forms (Figs. S3 & S4), it would be very

informative to compare these spectra to probe changes in the conformation of G-alpha in the heterotrimer.

We appreciate the reviewer's suggestion. We added the overlay of the ^1H - ^{13}C HMQC spectra of the free Gai3 and the Gai3 $\beta\gamma$ complex, the plot of the normalized chemical shift differences, and the mapping of the methyl groups with significant chemical shift differences onto the structure (Supplementary Figure 5). We observed significant chemical shift differences between the free Gai3 and its complex on the N-terminal region, the switch I region (residues 177-186), and the switch II region (residues 199-220), which all form the binding surface for G $\beta\gamma$. These results support the fact that the Gai3 forms the complex with G $\beta\gamma$ in our NMR sample, in a similar manner to that observed in the crystal structure of the G $\alpha\beta\gamma$ complex.

Regarding this point, we added an explanation of the chemical shift differences between the Gai3 alone and the Gai3 $\beta\gamma$ complex, as follows.

Supplementary Figure 5 **Chemical shift differences between Gai3(GDP) and Gai3 $\beta\gamma$.** (A) Overlay of the ^1H - ^{13}C HMQC spectra of Gai3 [ILVA] in the GDP-bound form (black) and Gai3[ILVA] $\beta\gamma$ (green). The methyl groups with significant chemical shift differences are labeled. (B) Plot of the chemical shift differences between Gai3(GDP) and Gai3 $\beta\gamma$. The averaged chemical shift differences are calculated by the equation, $\Delta\delta = [(\Delta\omega_{\text{H}}^2 + (\Delta\omega_{\text{C}}/5.9)^2)^{0.5}]$. The methyl groups with chemical shift differences are colored according to their $\Delta\delta$ values. (C) The methyl groups with significant chemical shift differences are mapped

on the crystal structure of Gai1 $\beta\gamma$ (PDB ID: 1GP2). Methyl groups are shown as spheres and colored according to their $\Delta\delta$ values. Source data are provided as a Source Data File.

(Pages 10-11, lines 212-220)

We focused on observing the G α subunit, since it confers the specificity, and prepared a selectively labeled {ul-[²H, ¹⁵N]; Ala β , Ile δ 1, Leu δ , Val γ -[¹³CH₃]} Gai3 complexed with [non-labeled] $\beta\gamma$ (Gai3[ILVA] $\beta\gamma$). We observed the ¹H-¹³C HMQC spectrum of Gai3[ILVA] $\beta\gamma$, and assigned the methyl signals based on the nuclear Overhauser effect spectroscopy and mutagenesis experiments (Supplementary Figure 4). By comparing the HMQC spectrum with that of Gai3 alone, which we previously reported ³², we confirmed the formation of the Gai3 $\beta\gamma$ complex that is consistent with the reported crystal structure ³³ (Supplementary Figure 5).

1-7) Another open question is whether the three constructs (in particular the chimeras) used to probe the specificity in the BRET assay show comparable nucleotide binding properties and GPCR coupling efficiency. Recent cryo-EM structures showed that not only the C-terminal 4 amino acid residues are required for GPCR interaction. Thus, the G-alpha-qi5 construct might not be activated by the chosen GPCRs.

We appreciate the reviewer's careful reading of our manuscript. The nucleotide binding properties and GPCR coupling efficiencies of the chimeric G α proteins were confirmed on the basis of the results from a set of positive control experiments, in which the agonist-dependent increase in BRET between Venus-G $\beta\gamma$ and GRK-Luc (Δ BRET_{GRK}) was observed. Supplementary Table 1 shows that the observed Δ BRET_{GRK} values were very similar between all of the chimeric G α proteins that were used to obtain the specificity factor (0.035 ~ 0.047 for the DOR-mediated activation, and 0.026 ~ 0.038 for the D2R-mediated activation), supporting our notion that the GPCR-induced activation of these chimeric G α proteins was not significantly perturbed by the chimerization. One exception is Gaiq5, which did not show a significant Δ BRET_{GRK} upon the addition of the DOR agonist (Δ BRET_{GRK} = 0.002), therefore, we did not use this affected pair (Gaiq5-DOR) to obtain the specificity factor to GIRK.

Supplementary Table 1 BRET values for all BRET experiments

			(1) Ligand free	(2) Met-Enkephalin	(3) Met-Enkephalin + 10 \times excess Naloxone	(2)-(1) Δ BRET	n
DOR + GRK-Luc	no G $\beta\gamma$	no G α	0.0877 \pm 0.0001	0.0858 \pm 0.0001	0.0837 \pm 0.0002	-0.0019	3
	Venus-G $\beta\gamma$	Gai3	0.1005 \pm 0.0023	0.1426 \pm 0.0053	0.1046 \pm 0.0030	0.0421	10
		Gaiqi	0.0899 \pm 0.0013	0.1246 \pm 0.0041	0.0904 \pm 0.0016	0.0347	10
		Gai3-q(α A)	0.0966 \pm 0.0021	0.1372 \pm 0.0046	0.0981 \pm 0.0019	0.0406	4
		Gai3-q(α B)	0.1026 \pm 0.0031	0.1456 \pm 0.0096	0.1053 \pm 0.0040	0.0430	3

		Gai3-q(α E)	0.1086 \pm 0.0038	0.1530 \pm 0.0065	0.1130 \pm 0.0046	0.0444	5
		* Gaiq5	0.0988	0.1008	0.0953	0.0020	1

			(1) Ligand free	(2) Met-Enkephalin	(3) Met-Enkephalin + 10 \times excess Naloxone	(2)-(1) Δ BRET	n
DOR +	no G $\beta\gamma$	no G α	0.0885 \pm 0.0001	0.0863 \pm 0.0001	0.0837 \pm 0.0002	-0.0022	3
GIRK-Luc	Venus-G $\beta\gamma$	Gai3	0.0900 \pm 0.0015	0.1020 \pm 0.0022	0.0917 \pm 0.0015	0.0120	10
		Gaiqi	0.0850 \pm 0.0010	0.0902 \pm 0.0015	0.0834 \pm 0.0010	0.0052	10
		Gai3-q(α A)	0.0917 \pm 0.0018	0.0982 \pm 0.0023	0.0887 \pm 0.0018	0.0065	4
		Gai3-q(α B)	0.0900 \pm 0.0007	0.1003 \pm 0.0014	0.0917 \pm 0.0008	0.0103	3
		Gai3-q(α E)	0.0948 \pm 0.0026	0.1057 \pm 0.0030	0.0961 \pm 0.0023	0.0109	5
		* Gaiq5	0.0874	0.0859	0.0844	-0.0015	1

			(1) Ligand free	(2) Met-Enkephalin	(3) Met-Enkephalin + 10 \times excess ICI-174,864	(2)-(1) Δ BRET	n
DOR +	no G $\beta\gamma$	no G α	0.0869 \pm 0.0004	0.0845 \pm 0.0004	0.0823 \pm 0.0004	-0.0024	3
GRK-Luc	Venus-G $\beta\gamma$	Gai3	0.1265 \pm 0.0083	0.1735 \pm 0.0055	0.1236 \pm 0.0072	0.0470	3
		Gaiqi	0.1096 \pm 0.0055	0.1540 \pm 0.0097	0.1082 \pm 0.0061	0.0444	3

			(1) Ligand free	(2) Met-Enkephalin	(3) Met-Enkephalin + 10 \times excess ICI-174,864	(2)-(1) Δ BRET	n
DOR +	no G $\beta\gamma$	no G α	0.0874 \pm 0.0008	0.0852 \pm 0.0008	0.0827 \pm 0.0005	-0.0022	3
GIRK-Luc	Venus-G $\beta\gamma$	Gai3	0.1006 \pm 0.0020	0.1101 \pm 0.0022	0.0987 \pm 0.0014	0.0095	3
		Gaiqi	0.0945 \pm 0.0009	0.1004 \pm 0.0022	0.0918 \pm 0.0013	0.0059	3

			(1) Ligand free	(2) Dopamine	(3) Dopamine + 10 \times excess Haloperidol	(2)-(1) Δ BRET	n
D2R +	no G $\beta\gamma$	no G α	0.0881 \pm 0.0003	0.0863 \pm 0.0001	0.0833 \pm 0.0003	-0.0018	3
GRK-Luc	Venus-G $\beta\gamma$	Gai3	0.1304 \pm 0.0035	0.1643 \pm 0.0057	0.1255 \pm 0.0029	0.0339	7
		Gaiqi	0.1215 \pm 0.0024	0.1487 \pm 0.0041	0.1192 \pm 0.0020	0.0272	7
		Gaiq5	0.1115 \pm 0.0009	0.1371 \pm 0.0032	0.1063 \pm 0.0009	0.0256	9
		Gaiqi5	0.1255 \pm 0.0019	0.1638 \pm 0.0039	0.1219 \pm 0.0018	0.0383	7

			(1) Ligand free	(2) Dopamine	(3) Dopamine + 10 \times excess Haloperidol	(2)-(1) Δ BRET	n
--	--	--	-----------------	--------------	---	-----------------------	---

D2R +	no Gβγ	no Gα	0.0885±0.0001	0.0863±0.0001	0.0834±0.0002	-0.0022	3
GIRK-Luc	Venus-Gβγ	Gai3	0.1046±0.0019	0.1147±0.0020	0.1005±0.0014	0.0101	7
		Gaiqi	0.1023±0.0021	0.1066±0.0021	0.0972±0.0016	0.0043	7
		Gaqi5	0.0968±0.0006	0.0993±0.0016	0.0913±0.0005	0.0025	9
		Gaqiqi5	0.1033±0.0009	0.1122±0.0009	0.1010±0.0007	0.0089	7

* indicates the conditions where the activation of G proteins was not observed.

To clarify this point, we added explanations of these positive control experiments, as follows.

(Pages 7-8, lines 140-150)

Rusinova and co-workers have previously reported that the helical domain of Gα confers the specificity for M2R-mediated GIRK activation¹³. Referring to this report, we compared the specificity factors when 3 different Gα proteins, Gai3, Gaqi5, and Gaiqi, were used (Figure 1E). Gai3 belongs to the i/o family of Gα and is responsible for GIRK activation in biological processes; Gaqi5 refers to Gaq with the C-terminal 5 residues replaced by those of Gai3 to couple with Gi/o-coupled GPCRs²⁶; and Gaiqi is a chimeric protein consisting of the GTPase domain of Gai3 (residues 1-62 and 176-354) and the helical domain of Gaq (residues 69-180)¹³. **We confirmed that all of the Gα chimeric proteins used to calculate the specificity factor showed similar ΔBRET_{GRK} values upon the addition of the GPCR agonists, indicating that they have comparable nucleotide binding properties and GPCR-coupling efficiencies (Supplementary Table 1).**

1-8) Also, the G-alpha-qi might show altered nucleotide binding properties and thus altered dissociation properties of the G-beta-gamma subunit.

We appreciate the reviewer's comment. We confirmed that Gaiqi exhibits similar nucleotide binding and Gβγ dissociation properties of to those of the native Gai3, on the basis of the following experimental results.

(i) We observed similar agonist-dependent increases in BRET between Venus-Gβγ and GRK-Luc (ΔBRET_{GRK}), in both experiments using Gai3 (0.0421±0.0032) and Gaiqi (0.0347±0.0031) (Figure 1B, Supplementary Table 1) . These results indicate that the levels of Venus-Gβγ, released in response to the GPCR-activation, are very similar between these two Gα proteins (please also refer to our reply to 1-7).

(ii) We confirmed that the binding affinities for Gβγ were not substantially different between Gai3 and Gaiqi, based on the competitive binding experiments between Gα and GRK-Luc. The results of the competitive binding experiments are summarized below. In the HEK293T cells expressing Venus-Gβγ and GRK-Luc, we additionally expressed Gai3 or Gaiqi by increasing the amounts of DNA encoding Gα. As the expression level of Gα increased, the basal BRET between Venus-Gβγ and GRK-Luc decreased, reflecting the phenomenon that the Gα competitively displaces

Venus-G $\beta\gamma$ from GRK-Luc. In the cases of both G α i3 and G α iqi, the BRET value decreased to a similar extent by increasing the amounts of DNA encoding G α . These results indicated that both G α i3 and G α iqi expressed at a sufficient level and formed a heterotrimer with G $\beta\gamma$ in the inactive state, and that the binding affinities for G $\beta\gamma$ were similar between these two G α proteins.

Supplementary Figure 2 BRET experiments observing the competitive binding of G α for Venus-G $\beta\gamma$. (A) Schematic representation of a competitive binding experiment observing the basal BRET between G $\beta\gamma$ and GRK. (B) Comparisons of the basal BRET values of cells expressing GRK-Luc, Venus-G $\beta\gamma$, and G α i3 (left) or G α iqi (right). When the G α expression levels were varied, the amounts of the transfected DNA encoding G α were varied between 0 and 570 ng, while the amounts of the transfected DNAs encoding GRK-Luc and Venus-G $\beta\gamma$ were fixed to 18 ng and 24 ng, respectively. (C) Schematic representation of a competitive binding experiment observing the basal BRET between G $\beta\gamma$ and GIRK. (D)

Comparisons of the basal BRET values of cells expressing GIRK-Luc, Venus-G $\beta\gamma$, and G α i3 (left) or G α iqi (right). When the G α expression levels were varied, the amounts of the transfected DNA encoding G α were varied between 0 and 570 ng, while the amounts of the transfected DNAs encoding GIRK1, GIRK2-Luc, and Venus-G $\beta\gamma$ were fixed to 18 ng, 18 ng, and 24 ng, respectively. Data are means \pm SEM of 3 measurements taken from independently transfected cell batches. Source data are provided as a Source Data File.

We added the explanations for the competitive binding experiments as follows.

(Page 8, lines 147-158)

We confirmed that all of the G α chimeric proteins used to calculate specificity factor showed similar Δ BRET_{GIRK} values upon the addition of the GPCR agonists, indicating that they show comparable nucleotide binding properties and GPCR-coupling efficiency (Supplementary Table 1). We **also conducted competitive binding experiments, in which we monitored the decrease in the BRET signal caused by the displacement of Venus-G $\beta\gamma$ bound to GRK-Luc, by increasing the amounts of G α . In the cases of both G α i3 and G α iqi, the BRET signal decreased to a similar extent by increasing the amounts DNA encoding G α , indicating that the replacement of the helical domain does not result in marked differences in the G $\beta\gamma$ -binding property in the inactive state (Supplementary Figure 2A, B). Similar results were obtained when we expressed GIRK-Luc and monitored the Δ BRET_{GIRK} values (Supplementary Figure 2C, D).**

1-9) The obtained structural model is based on PRE experiments with three single cysteine variants shows that the main interaction is mediated by the C-terminal tail of GIRK. The large dynamics of the complex is accounted for by using structure ensembles for the back-calculation of the PRE effects. This yields a structural ensemble exhibiting still very large changes in the orientation of the individual members (Fig. S6), limiting the value of such a sophisticated calculation. In the end, the question is how relevant the three poses shown in Fig S7 are for the mechanism of GIRK activation.

We appreciate the reviewer's comment. The functional significance of the large dynamics of the G $\alpha\beta\gamma$ -GIRK complex is that the interaction between G $\beta\gamma$ and GIRK, followed by the activation of the GPCRs, is not sterically perturbed by the nearby G $\alpha\beta\gamma$, and that the rapid activation of GIRK can be achieved while retaining the G α -family specificity. This is because the G $\beta\gamma$ -binding site can be transiently vacant, thanks to the inherent flexibility of the G $\alpha\beta\gamma$ orientation relative to GIRK.

We note that the large variations in the orientation of G $\alpha\beta\gamma$ do not necessarily mean that the ensemble calculation results are obscured, but the dynamic complex is a mechanical description of the physiologically relevant G $\alpha\beta\gamma$ -GIRK interaction. This notion is supported by the fact that the experimentally observed PREs from L366C-TEMPO cannot be well-fitted by a single structure, and that we have to conduct ensemble-structural calculations to fully explain the experimentally

observed PREs.

To clarify these points, we added explanations for the flexibility of the relative orientation and the functional significance of the multiple poses. We also changed the Fig. S7 (Supplementary Figure 9 in the revised manuscript) to include the schematic explanation of the rapid activation of GIRK.

(Pages 14-15, lines 307-313)

We sought to visualize the structure of the $G\alpha i3\beta\gamma$ -GIRK complex by structural calculation using the observed PREs as distance restraints. However, our initial attempt to obtain a single complex structure that simultaneously satisfies the PRE patterns from Q344C-, V351C-, and L366C-TEMPO failed, as indicated by the relatively large Q-factor³⁸ of 0.71, even in the best fit result. **This result indicates that the relative orientation between $G\alpha i3\beta\gamma$ and GIRK in the complex is inherently flexible, and we must use an ensemble of structures to explain the observed PREs.**

(Pages 19-20 in the supplementary information)

The variations in the relative orientations of $G\alpha i/o\beta\gamma$ are in contrast to the fixed orientation of $G\beta\gamma$ toward GIRK, as demonstrated in our NMR analyses of the $G\beta\gamma$ -GIRK interaction and the crystal structure of the $G\beta\gamma$ -GIRK complex (Whorton and MacKinnon, 2013; Yokogawa et al., 2011). Since the $G\beta\gamma$ -binding site on GIRK is located on the structured β -strand region, the flexibility in the relative orientation of $G\alpha i/o\beta\gamma$ allow $G\beta\gamma$ to access GIRK without being hindered by $G\alpha i/o\beta\gamma$, **by making the $G\beta\gamma$ -binding site on GIRK transiently vacant.** Thus, we assume that the inherent flexibility in the relative orientation plays important roles in achieving the rapid activation of GIRK (Supplementary Figure 9).

(Supplementary Figure 9)

Supplementary Figure 9 Representative poses of the membrane-directed $G\alpha i3\beta\gamma$ in complex with the GIRK chimera. (A) Side and top views of one structure (Pose 1), along with top views of two other structures (Poses 2 and 3) are displayed. The direction of the N-terminal helix in the pose 1 is marked as a red dotted line, and the relative angles of $G\alpha i3\beta\gamma$ in Poses 2 and 3 to that in Pose 1 are shown. **The open $G\beta\gamma$ -binding site in the Pose 3 orientation is shown as a blue dotted circle.** (B) Schematic representation of the rapid activation of GIRK from the $G\alpha i3\beta\gamma$ -GIRK complex. **In the $G\alpha i3\beta\gamma$ -GIRK complex, the relative orientation of $G\alpha i3\beta\gamma$ toward the β -strand region of GIRK is rather flexible and the $G\beta\gamma$ -binding site on GIRK is transiently vacant, which enables the rapid activation of GIRK upon the agonist-binding to GPCR.**

1-10) Furthermore, how was the relative orientation of the N-terminal G-alpha helix and G-beta defined? These features cannot be obtained with the described PRE experiments where only G-alpha is labeled with isotopes.

We appreciate the reviewer's comment. In our ensemble calculations, $G\alpha\beta\gamma$ is treated as a stable complex, in which the relative orientation between $G\alpha$ and $G\beta\gamma$ is fixed. Therefore, if we can determine the position and orientation of $G\alpha$ using the distance information obtained from the PREs

on $G\alpha$, we can simultaneously determine those of $G\beta\gamma$. The assumption of the stable complex is made on the basis of the fact that the interaction between $G\alpha$ and $G\beta\gamma$ in the inactive state is very stable, with a K_d on the order of 1 nM (Sarvazyan et al., 1998), and $G\alpha$ binds to $G\beta\gamma$ with extensive contacts larger than 1,300 Å² in the complex.

To clarify this point, we added an explanation of the assumption of the stable $G\alpha\beta\gamma$ complex in the revised manuscript as follows.

(Pages 29-30, lines 668-673)

Five hundred complex structures were generated, and the lowest energy structure was selected to be restored into the full-length GIRK chimera-nanodisc and $G\alpha_i3\beta\gamma$ by superposition. **In the following calculations, $G\alpha_i3\beta\gamma$ was treated as a stable complex, so the orientation and position of $G\beta\gamma$ could be simultaneously determined on the basis of the distance information observed for $G\alpha_i3$.**

1-11) The paper would also benefit from a more thorough discussion on other complexes formed by G-alpha and G-beta-gamma upon activation by a GPCR. How likely is an association with GIRK considering the presence of partner proteins like adenylylase and others? How could this be integrated into the model shown in Fig. 5?

We appreciate the reviewer's suggestion. To the best of our knowledge, the co-localizations between GIRK and other effector proteins, such as adenylylase, have not been reported so far. Therefore, the association of GIRK with G proteins, as shown in Fig.5, does not seem to be affected in the presence of other effector proteins. We expect that the activation of the other effector proteins is induced by G proteins that do not participate in the GIRK- $G\alpha\beta\gamma$ complex.

To clarify this point, we added a discussion about the possible effects on the other effector proteins, as follows.

(Page 19, lines 414-422)

These structural models suggest that the stepwise changes in the interaction modes between GIRK and G proteins enable the specific and efficient regulation of GIRK in response to extracellular stimuli, which would play critical roles in the robust and selective signal transductions in the heart and neural systems (Please also see Supplementary Discussion). **We expect that the regulation of other effector proteins, such as adenylylase, is not affected by the formation of the $G\alpha_i/o\beta\gamma$ -GIRK complex, because these effectors do not co-localize with GIRK and can be independently regulated by G proteins that do not participate into the $G\alpha_i/o\beta\gamma$ -GIRK complex.**

Reviewer 2

2-1) Nanodiscs provide a lipid environment, which can also incorporate hydrophobic entities such as e.g. spin labels. This raises concerns with regard to the interpretation of PRE measurements. Can the authors

convincingly show that there was no spin label contamination into the nanodiscs and that the measured contributions to PRE are the specific result of TEMPO attached to GIRK?

We appreciate the reviewer's comment. We confirmed that the observed PREs are not from spin label contamination into the nanodiscs, on the basis of the following results.

(i) If the spin label contamination into the nanodiscs contributes to the observed PREs, then these non-specific PREs are expected to be observed on similar methyl groups and with similar magnitudes, regardless of the spin-labeled sites on GIRK. In our experiments, the methyl groups with marked PREs are distributed quite differently, depending on the spin-labeled sites on GIRK (Figure 3), indicating that the non-specific PREs are not observed in our experiments.

(ii) We conducted experiments observing PREs from spin-labeled molecules that partition into the membrane, 4-palmitamido-TEMPO. When we added the 4-palmitamido-TEMPO at 500 μM in the presence of GIRK-nanodiscs, we were not able to observe significant PREs from 4-palmitamido-TEMPO on $\text{G}\alpha\text{i}3\beta\gamma$. This result indicates that the PREs from the spin-labeled molecules partitioned into the nanodisc membrane do not result in large PREs, probably due to the high mobility of the spin labeled molecules inside the membrane. This observation is consistent with the previous report that the PREs observed from spin-labeled molecules distributed into nanodiscs are relatively small, even if the observed proteins strongly bind to the nanodisc membrane with K_d values on the order of 10^2 - 10^3 nM (Yokogawa et al., 2012).

These results indicate that the PREs from spin label contamination into the nanodiscs minimally contribute to the observed PREs under our experimental conditions.

A 4-palmitamido-TEMPO

B $\text{G}\alpha\text{i}3[\text{ILVA}]\beta\gamma$ 20 μM + GIRK chimera-nanodiscs 30 μM
+ 4-palmitamido-TEMPO 500 μM

Supporting Figure 3 PRE experiments using spin-labeled molecules that distribute into the nanodisc membrane. (A) Chemical structure of 4-palmitamido-TEMPO. (B) Plots of Γ_{PRE} observed on $\text{G}\alpha\text{i}3[\text{ILVA}]\beta\gamma$ from 4-palmitamido-TEMPO in the presence of the GIRK chimera-nanodiscs.

2-2) *The authors are using a BRET assay to determine the specificity of different Ga domains in activating*

GIRK. The outcome of two experiments is compared: one which provides the $G\alpha$ specific GIRK binding to $G\beta\gamma$ while the other one gives the GPCR related signaling level due to $\beta\gamma$ interaction with a GRK. These measurements are reflecting the thermodynamic stability of the interactions and accordingly thermodynamic control. However, as fast switching of the GIRK seems to be a feature of the mechanism it could be that the channel activation through $G\beta\gamma$ is under kinetic control and hence not related to the BRET values measured. The authors should comment on this possibility.

We appreciate the reviewer's comment. We concluded that the specificity in the GIRK activation is under thermodynamic control, on the basis of the following observations.

(i) In electrophysiological analyses of GIRK, the $G\alpha i/o$ specificity in activating GIRK was mainly observed in the differences in the steady-state GIRK current upon the addition of the GPCR agonist, and not in the activation kinetics of GIRK.

[redacted]

Supporting Figure 4 Electrophysiological recordings of GIRK reported by Rusinova and co-workers (Fig.1 from (Rusinova et al., 2007)). (A) The trace of the GIRK current upon the activation of $G\alpha i3$. (B) The trace of the GIRK current upon the activation of $G\alpha q i5$. The GTPase and helical domains of $G\alpha q i5$ are from and Gq , and the C-terminal five residues are replaced with those of $G\alpha i3$. $G\alpha q i5$ can be activated by the Gi-coupled M2 ACh receptor.

(ii) Touhara and MacKinnon have recently demonstrated that GIRK exists in a fast equilibrium between the free- and $G\beta\gamma$ -bound states, and the magnitude of the GIRK current can be explained by the fraction of the $G\beta\gamma$ -bound state ($[GIRK-\beta\gamma_4]$) in the equilibrium (i.e., the thermodynamic stability of the $G\beta\gamma$ -bound state) (Touhara and MacKinnon, 2018).

Supporting Figure 5 (A) Reaction scheme used to model the GPCR activation of GIRK, proposed by Touhara and MacKinnon (Fig.7A from Touhara and MacKinnon, *eLife* 2018). The extent of the GIRK activation can be explained by the population of the $G\beta\gamma$ -bound state ([GIRK- $\beta\gamma_4$]). (B) The simulations of the ACh-stimulated GIRK current (Fig.7B from Touhara and MacKinnon, *eLife* 2018). It only took about 2-3 seconds to reach equilibrium. The GIRK currents from two different SAN cells are shown in grey. Calculated [GIRK- $\beta\gamma_4$] concentrations as a function of time for two different k_{12} magnitudes are shown by black solid and dashed curves, where k_{12} denotes the association rate constant between $G\alpha\beta\gamma$ and GPCR.

(iii) In our time-resolved BRET measurements, we observed similar activation rates between Gi3 and Giq1, showing that the activation kinetics were not significantly different between them.

These lines of evidence support the hypothesis that the activation of GIRK is under thermodynamic control, and the difference in the activation kinetics is not the main reason for the $G\alpha$ -specificity. Therefore, we can determine the specificity of $G\alpha$ in activating GIRK by using a BRET assay, which monitors the thermodynamic stability of the interactions.

To comment on this point, we added the following explanation to clarify that the *Gai/o* specificity in activating GIRK is under thermodynamic control.

(Page 6, lines 97-109)

First, we conducted cell-based assays to quantitatively evaluate the *Gai/o*-specific activation of GIRK, to

identify the structural element of $G\alpha$ that determines the specificity in the activation of GIRK. To date, the $G\alpha i/o$ specificity in regulating GIRK has been mainly characterized by electrophysiological analyses observing GPCR agonist-induced GIRK currents, using cultured cells or oocytes expressing a GPCR, GIRK and various $G\alpha$ mutants^{10,13}. In these experiments, **the $G\alpha i/o$ specificity in activating GIRK was mainly observed as the differences in the steady-state GIRK current upon the addition of GPCR agonists, indicating that the preference of $G\alpha i/o$ is mainly under thermodynamic control, rather than kinetic control. This steady-state GIRK-current has been compared between $G\alpha$ families to characterize the $G\alpha i/o$ specificity, however,** the observed GIRK currents are strongly affected by the extents of G protein activation; i.e., the amounts of $G\beta\gamma$ released upon the activation of GPCRs, which can significantly differ among the $G\alpha$ mutants analyzed.

2-3) The NMR investigations are conducted with GIRK embedded in a nanodisc environment. This allows to approximate various effects that might be related to protein encounters in the membrane proximity and to mimic influence related to the $G\alpha$ N-terminal interaction with the bilayer. This all makes sense and is a good choice in order to project the results of the in vitro investigations into a physiological environment. However, typically NMR work uses rather small nanodiscs to reduce detrimental effects related to fast spin relaxation. I simply wonder about the geometric constraints here. It would be useful if the authors could provide some information on the physical size of the nanodisc diameter and the size of the interacting proteins, so that one can conclude on the physical size requirements of the complex and assess whether any interference from the nanodisc scaffold protein might have influenced the study.

We appreciate the reviewer's suggestion. The diameter of the nanodisc composed of MSP1E3 is estimated to be around 120 Å (Denisov et al., 2004). The diameter of GIRK is estimated to be around 75 Å at the cytoplasmic region and 45 Å at the transmembrane region, and the length of the N-terminal helix of $G\alpha\beta\gamma$ is estimated to be around 40 Å. Therefore, the GIRK- $G\alpha\beta\gamma$ complex is expected to fit well within the nanodisc, without interference from the scaffold protein.

Supporting Figure 6 Model structure of the GIRK- $G\alpha\beta\gamma$ complex in an MSP1E3 nanodisc. The

model structure of the G α i3 β γ -GIRK chimera complex is the same as the structure shown in Fig. 4.

To comment on this point, we added an explanation of the size of the nanodisc, as follows.

(Page 24, lines 533-542)

The GIRK chimera protein was solubilized in 20 mM of *n*-dodecyl- β -D-maltoside (Dojindo) and purified by chromatography on HIS-Select Nickel Affinity Gel. For nanodisc reconstitution, a lipid mixture comprising 70% 1-palmitoyl-2-oleoyl-phosphatidylcholine, 25% 1-palmitoyl-2-oleoyl-phosphatidylglycerol, and 5% L- α -phosphatidylinositol-4,5-bisphosphate (Brain, Porcine) (w/w) (Avanti Polar Lipid) was desiccated and dissolved in 50 mM sodium cholate. We used nanodiscs composed of MSP1E3 with an approximate diameter of 120 Å⁴⁹, which is sufficiently large to accommodate the complex of G α i/ β γ with about a 40 Å length at the lipid-binding N-terminal region, and the GIRK chimera with a diameter of 45 Å at the transmembrane region.

2-4) Figure 2C shows the reduction in intensity rather than a simple intensity ratio, which is typically shown in such situations. It might be less confusing to show the latter and the figure and text should be adapted accordingly.

We appreciate the reviewer's comment. We showed the simple intensity ratios upon the addition of GIRK-nanodiscs in Fig. 2, as follows.

Figure 2 NMR spectral changes of Gai3[ILVA] $\beta\gamma$ induced by the GIRK chimera-nanodiscs. (A) Overlay of the ^1H - ^{13}C HMQC spectra of Gai3[ILVA] $\beta\gamma$ in the presence (red) and absence (black) of 2 eq. of the GIRK chimera-nanodiscs. (B) Close-up views and cross-sections of V13 γ 2, L148 δ 1, and L310 δ 2 as representative signals. For comparison, the corresponding signals of Gaiqi[ILVA] $\beta\gamma$ in the presence (blue) and absence (black) of the GIRK chimera-nanodiscs are shown on the right. (C) Plots of relative intensities of Gai3[ILVA] $\beta\gamma$ (top) and Gaiqi[ILVA] $\beta\gamma$ (middle) in the presence of 2 eq. of the GIRK chimera-nanodiscs. **A plot of the relative intensities of Gai3[ILVA] $\beta\gamma$ in the presence of empty nanodiscs is also shown (bottom).** The error bars are calculated based on the signal-to-noise ratios. Methyl groups with relative intensities lower than 0.80 are colored according to their values. (D) Mapping of the methyl groups of Gai3 $\beta\gamma$ with significant intensity reductions on the structure of Gai1 $\beta\gamma$ (PDB ID: 1GP2)³³. Methyl groups are shown as spheres and colored according to their relative intensity values. **Source data are provided as a Source Data File.**

2-5-1) On several occasions there is a relatively diffuse mention of Gai/o N-terminal interaction with the lipid bilayer, which to some extent mimics the anchoring of the heterotrimeric G protein to the membrane. The authors could not closely simulate the full anchoring taking place in the cellular environment as they found that the corresponding lipidated G protein heterotrimer aggregated. Hence, left only with the N-terminus supposedly interacting with the bilayer one has to assume that there is sufficient affinity for Ga to 'stick' to the nanodisc lipid bilayer. There is no quantification provided.

We appreciate the reviewer's suggestion. In order to further prove that the N-terminal region of Ga has sufficient affinity for the membrane and to quantify this contribution, we conducted intermolecular PRE experiments using GIRK solubilized in detergent micelles, and compared the results with those obtained with GIRK-nanodiscs.

Although we were able to observe similar PREs on the helical domain of Gai3 $\beta\gamma$ from L366C-TEMPO of GIRK in DDM micelles, we had to add GIRK at a 2-fold higher concentration (50 μM) to observe similar magnitudes of PREs, as compared to the GIRK-nanodiscs (25 μM). On the basis of these results, we estimated that the interaction between Ga and the nanodisc membrane enhances the affinity of Ga $\beta\gamma$ for GIRK by about 2-fold. This quantification further supports our notion that the N-terminal region of Ga interacts with the membrane and significantly contributes to the formation of the Ga $\beta\gamma$ -GIRK complex.

Supporting Figure 7 (Top) Plots of the intermolecular PREs on Gai3[ILVA]βγ from L366C-TEMPO of GIRK solubilized in DDM micelles. The concentration of GIRK was 50 μM , as a tetramer. (Bottom) Plots of the intermolecular PREs on Gai3[ILVA]βγ from L366C-TEMPO of GIRK in nanodiscs. The concentration of the GIRK-nanodiscs was 25 μM , as a particle. The signals with significant PREs ($\Gamma_2^{\text{obs}} > 7 \text{ s}^{-1}$) from L366C-TEMPO are labeled. The methyl groups with significant PREs are mapped on the structure.

2-5-2) Based on that, it is not immediately obvious to me why the intensity comparison was conducted with reference to the freely soluble Gai3 domain, in absence of nanodiscs. I would have thought that comparing it to the intensities in the presence of nanodiscs would provide a better reference for signal intensities. Using the latter would improve the confidence in the observed intensity changes upon GIRK binding. Likewise, it would also allow a better estimate of the K_d , currently put at 200 μM .

We appreciate the reviewer's suggestion. According to the reviewer's suggestion, we conducted a control experiment using empty nanodiscs. Upon the addition of the empty nanodiscs, we did not observe significant intensity reductions of Gai3[ILVA]βγ. This result indicates that the interaction with the membrane lipid alone is not strong enough to facilitate the formation of the complex, and that the population of Gai3[ILVA]βγ bound to the empty nanodiscs is so small that we could not detect the NMR spectral changes. This result also indicates that the marked intensity reductions observed upon the addition of GIRK-nanodiscs mainly reflects the contribution from the binding of Gai3[ILVA]βγ to GIRK, therefore, our estimation of K_d would not be largely affected by this result.

As we demonstrated in our reply to 2-5-1, the interaction with the nanodisc membrane also significantly contributes to the complex formation. These results support our notions that Gai3βγ needs to interact with both GIRK and the nanodisc membrane to stably form the Gai3βγ-GIRK complex, and that the intensity reductions observed in Gai3[ILVA]βγ are mainly caused by the

specific interaction between $G\alpha\beta\gamma$ and GIRK. (Please refer to the reply to point 1-4 raised by Reviewer 1)

In the revised manuscript, we added the results of the control experiment with the empty nanodiscs.

(Page 11, lines 220-234)

Upon the addition of 2 equivalents of the GIRK chimera-nanodiscs to $G\alpha\beta\gamma$ [ILVA], most of the signals exhibited intensity reductions with relative intensities lower than 0.9, and the signals from L5 δ 1, L5 δ 2, A12 β , V13 γ 1, V13 γ 2, A30 β , A31 β (N-terminal helix), L36 δ 1, L36 δ 2, L37 δ 1, L37 δ 2 (β 1 strand), A41 β (β 1- α 1 loop), I127 δ 1 (α C helix), L148 δ 1 (α D- α E loop), L159 δ 2 (α E helix), V218 γ 2 (α 2- β 4 loop), I221 δ 1 (β 4 strand), L232 δ 2, L234 δ 2 (β 4- α 3 loop), L249 δ 1, L249 δ 2, I253 δ 1 (α 3 helix), I264 δ 1, I265 δ 1 (β 5 strand), I278 δ 1, L283 δ 1 (α G- α 4 loop), and L348 δ 1 (C-terminus) exhibited further reduced intensities lower than 0.8 (Figure 2A-C), while the observed chemical shift changes were very small (< 0.01 ppm). **When we added the empty nanodiscs, we did not observe significant intensity reductions, showing that the observed intensity reductions upon the addition of the GIRK chimera-nanodiscs are mainly triggered by the specific binding of $G\alpha\beta\gamma$ to the GIRK chimera (Figure 2C).** The overall intensity reductions are caused by slower tumbling, due to an increased average molecular weight, indicating that a fraction of $G\alpha\beta\gamma$ forms a complex with the GIRK chimera-nanodiscs.

Figure 2 NMR spectral changes of $G\alpha\beta\gamma$ [ILVA] induced by the GIRK chimera-nanodiscs. (A)

Overlay of the ^1H - ^{13}C HMQC spectra of Gai3[ILVA] $\beta\gamma$ in the presence (red) and absence (black) of 2 eq. of the GIRK chimera-nanodiscs. (B) Close-up views and cross-sections of V13 γ 2, L148 δ 1, and L310 δ 2 as representative signals. For comparison, the corresponding signals of Gaiqi[ILVA] $\beta\gamma$ in the presence (blue) and absence (black) of the GIRK chimera-nanodiscs are shown on the right. (C) Plots of relative intensities of Gai3[ILVA] $\beta\gamma$ (top) and Gaiqi[ILVA] $\beta\gamma$ (middle) in the presence of 2 eq. of the GIRK chimera-nanodiscs. **A plot of the relative intensities of Gai3[ILVA] $\beta\gamma$ in the presence of empty nanodiscs is also shown (bottom).** The error bars are calculated based on the signal-to-noise ratios. Methyl groups with relative intensities lower than 0.80 are colored according to their values. (D) Mapping of the methyl groups of Gai3 $\beta\gamma$ with significant intensity reductions on the structure of Gai1 $\beta\gamma$ (PDB ID: 1GP2)³³. Methyl groups are shown as spheres and colored according to their relative intensity values. **Source data are provided as a Source Data File.**

2-6-1) The chemical shift changes shown in Fig 2 are with < 0.01 ppm very small. Based on these small changes, how is it possible to get further intensity reductions due to differential line broadening, as the authors claim. On p.22-23 it is argued that some of the signal intensity drops observed for interacting residues are related to differential line broadening, postulating that exchange is on the slow to intermediate side of coalescence.

We appreciate the reviewer's comment. The observed intensity reductions can be reasonably explained from the differential line broadening effects, based on the experimentally derived parameters. The detailed explanations are given as follows.

(i) The observed chemical shift difference, ~ 0.007 ppm, reflects the population-weighted average in the fast-exchange regime. Therefore, if we assume that the bound fraction is about 0.1, then the chemical shift difference between the free and GIRK-bound states can be estimated to be about 0.07 ppm.

(ii) The exchange rate between the free- and bound- states is estimated to be about $2,200 \text{ s}^{-1}$, under the experimental conditions of the binding experiment, if we assume the following parameters: $K_D = 200 \text{ }\mu\text{M}$, $k_{on} = 10^7 \text{ M}^{-1}\text{s}^{-1}$ (diffusion-limited), $[\text{GIRK-nanodisc}] = 22 \text{ }\mu\text{M}$, and $[\text{G}\alpha\beta\gamma] = 11 \text{ }\mu\text{M}$. Under these conditions, the exchange contribution (R_{ex}) in ^1H R_2 can be estimated to be 2.7 s^{-1} at 14.1 Tesla (600 MHz ^1H frequency).

(iii) The intensity reduction caused by the ^1H exchange broadening effects in the HMQC spectrum can be formulated as $\exp(-R_{ex}t_{\text{HMQC}}) * R_{\text{HC}} * R_{\text{H}} * (R_{\text{HC}} + R_{ex})^{-1} * (R_{\text{H}} + R_{ex})^{-1}$; where t_{HMQC} denotes the magnetization transfer time (6.9 ms), and R_{HC} and R_{H} denote the intrinsic relaxation rates in the F1 and F2 directions, respectively. The R_{HC} and R_{H} rates of free G $\alpha\beta\gamma$ are distributed around $20\text{-}30 \text{ s}^{-1}$.

(iv) By using these parameters, the intensity reduction is expected to be 0.12-0.15, which is consistent with the extent of the experimentally-observed differential line broadening.

Regarding these points, we added the estimations of the extent of differential line-broadening in the main text, as follows.

(Page 12, lines 243-252)

Assuming that the overall intensity reduction (~ 0.1) reflects the apparent increase in the molecular weight as a function of the bound population, we estimated the apparent K_d to be larger than 200 μM . The residues with significant intensity reductions were located on the N- and C-terminal regions, the $\text{G}\beta\gamma$ -binding site within the GTPase domain, and the helical domain of $\text{G}\alpha\text{i}3$ (Figure 2D), suggesting that these regions exhibit chemical shift differences caused by the direct contact with the GIRK chimera-nanodiscs, and/or by the conformational changes that occur upon the interaction. **This estimation of the K_d value is also consistent with the site-specific intensity reductions that were as large as 0.3, if we assume that the on-rate is diffusion limited ($k_{\text{on}} \sim 10^7 \text{ M}^{-1} \text{ s}^{-1}$) and the ^1H chemical shift difference between the free- and bound-states is around 0.05-0.1 ppm.**

2-6-2) On what experimental evidence is the latter assumption residing? The chemical shift changes shown in Fig. 2 are very small (quote < 0.01 ppm), hence, very likely the exchange should be on the fast exchange side of signal coalescence. I notice that two spectrometer field strengths were used in the study. Has field dependence been noticed validating the author's assumptions?

We appreciate the reviewer's careful reading of our manuscript. In the specific case of the GIRK- $\text{G}\alpha\beta\gamma$ interaction, it is appropriate to assume that the exchange between the free- and bound-states is in the intermediate-to-fast regime, on the basis of the above calculation.

Regarding this assumption, we corrected the sentence as follows.

(Page 11, lines 231-238)

The overall intensity reductions are caused by slower tumbling, due to an increased average molecular weight, indicating that a fraction of $\text{G}\alpha\text{i}3[\text{ILVA}]\beta\gamma$ forms a complex with the GIRK chimera-nanodiscs. The further intensity reductions are caused by differential line broadening, which results from the chemical shift changes in an **intermediate-to-fast** exchange regime between the free and the bound states, and/or the effect of the anisotropic tumbling induced by the binding, although the effect of the anisotropic tumbling was estimated to be relatively small for the membrane proteins in nanodiscs³⁴.

As the reviewer suggested, the measurement of the magnetic field dependence is an attractive approach to validate the existence of the exchange broadening contributions. However, in our cases, the R_{ex} contribution is smaller than 3 s^{-1} : thus the magnetic field- dependent changes in R_{ex} are expected to be a few s^{-1} . Unfortunately, these changes are too small to quantitatively characterize.

2-6-3) An alternative explanation for the increased intensity drops could be related to anisotropic tumbling accentuated in the bound state. Such anisotropic tumbling should already be visible in the nanodisc-associated state of the heterotrimeric G protein as facilitated via the N-terminal interaction of the Gai with the lipid bilayer. This seems to be another good reason why using the latter state as a reference for the intensity comparison

should be a better choice (see point 5).

We appreciate the reviewer's comment. We admit that the intensity reductions can also be induced by the accentuated anisotropic tumbling in the bound state, and thus we added their possible contributions in the main text.

We expect that the contributions of the anisotropic tumbling are relatively small as compared to those of the exchange broadening, on the basis of the previous study that investigated the relaxation properties of the OmpX protein reconstituted into nanodiscs. In the case of the OmpX protein in MSPΔH5 nanodiscs, the elevated R_2 rate could mainly be explained from the exchange broadening effect, and not from the effect of the anisotropic tumbling (Frey et al., 2017).

As we mentioned in our response to 2-5-2, we did not observe the significant reductions in the intensity of Gαi3βγ upon the addition of the empty nanodiscs. Therefore, the marked intensity reductions were mainly attributed to the interaction formed between Gαi3[ILVA]βγ and GIRK.

We added explanations of the possible contributions of the anisotropic tumbling in the main text, as follows.

(Page 11, lines 234-238)

The further intensity reductions are caused by differential line broadening, which results from the chemical shift changes in an intermediate-to-fast exchange regime between the free and the bound states, **and/or the effect of the anisotropic tumbling induced by the binding, although the effect of the anisotropic tumbling was estimated to be relatively small for membrane proteins in nanodiscs³⁴.**

Reviewer 3

Major

The authors use a BRET assay monitoring Gβγ/GIRK interaction as a surrogate for directly measuring K+ currents. This allows them to normalize GIRK activation to G protein activation, which is measured by GβγGRK BRET. This normalization is appealing but may also be problematic.

3-1) For example, it is unclear whether Gβγ binding to GRKs does not exhibit some Gα subunit preference, similar as the GIRKs do.

We appreciate the reviewer's careful reading of our manuscript. The GRK used in our study, masGRK3ct, only contains the Gβγ-binding domain (residues 495-688), and does not contain any regions that bind to Gα (around residues 100-140). The previously reported photobleaching experiments revealed that masGRK3ct does not form a pre-assembled complex with G proteins (Hollins et al., 2009). Therefore, masGRK3ct does not exhibit Gα preference, and has been widely used as a reporter for free Gβγ (Masuho et al., 2015; Maziarz and Garcia-Marcos, 2017).

We added an explanation that masGRK3ct does not exhibit a Gα family preference, and also added the references that support this fact.

(Page 6, lines 112-118)

We conducted two sets of BRET experiments for each G α protein: one to observe the intermolecular BRET between G $\beta\gamma$ and GIRK that reflects the GPCR-mediated GIRK activation, and the other to observe the intermolecular BRET between G $\beta\gamma$ and the G $\beta\gamma$ -binding domain of G protein-coupled receptor kinase (hereafter referred to as GRK), **which does not exhibit G α family preference** and serves as a reporter of the G protein activation²³⁻²⁵.

3-2) Moreover, since according to the authors already the heterotrimeric G protein assembles with GIRKs (the schemes in Figure 1A, C are in that sense misleading, compare with Figure 5a) it is important to know how much basal G $\beta\gamma$ /GIRK BRET is obtained with the different G proteins used. One would expect a higher basal BRET for G $\beta\gamma$ /GIRK with G proteins that bind GIRKs. Therefore, the basal BRET before agonist stimulation should be provided for all BRET experiments.

We appreciate the reviewer's suggestion. We added a supplementary table that includes the basal BRET values for all BRET experiments.

Supplementary Table 1 BRET values for all BRET experiments

			(1) Ligand free	(2) Met-Enkephalin	(3) Met-Enkephalin + 10 \times excess Naloxone	(2)-(1) Δ BRET	n
DOR + GRK-Luc	no G $\beta\gamma$	no G α	0.0877 \pm 0.0001	0.0858 \pm 0.0001	0.0837 \pm 0.0002	-0.0019	3
	Venus-G $\beta\gamma$	Gai3	0.1005 \pm 0.0023	0.1426 \pm 0.0053	0.1046 \pm 0.0030	0.0421	10
		Gaiqi	0.0899 \pm 0.0013	0.1246 \pm 0.0041	0.0904 \pm 0.0016	0.0347	10
		Gai3-q(α A)	0.0966 \pm 0.0021	0.1372 \pm 0.0046	0.0981 \pm 0.0019	0.0406	4
		Gai3-q(α B)	0.1026 \pm 0.0031	0.1456 \pm 0.0096	0.1053 \pm 0.0040	0.0430	3
		Gai3-q(α E)	0.1086 \pm 0.0038	0.1530 \pm 0.0065	0.1130 \pm 0.0046	0.0444	5
		* Gaqi5	0.0988	0.1008	0.0953	0.0020	1

			(1) Ligand free	(2) Met-Enkephalin	(3) Met-Enkephalin + 10 \times excess Naloxone	(2)-(1) Δ BRET	n
DOR + GIRK-Luc	no G $\beta\gamma$	no G α	0.0885 \pm 0.0001	0.0863 \pm 0.0001	0.0837 \pm 0.0002	-0.0022	3
	Venus-G $\beta\gamma$	Gai3	0.0900 \pm 0.0015	0.1020 \pm 0.0022	0.0917 \pm 0.0015	0.0120	10
		Gaiqi	0.0850 \pm 0.0010	0.0902 \pm 0.0015	0.0834 \pm 0.0010	0.0052	10
		Gai3-q(α A)	0.0917 \pm 0.0018	0.0982 \pm 0.0023	0.0887 \pm 0.0018	0.0065	4
		Gai3-q(α B)	0.0900 \pm 0.0007	0.1003 \pm 0.0014	0.0917 \pm 0.0008	0.0103	3
		Gai3-q(α E)	0.0948 \pm 0.0026	0.1057 \pm 0.0030	0.0961 \pm 0.0023	0.0109	5
		* Gaqi5	0.0874	0.0859	0.0844	-0.0015	1

			(1) Ligand free	(2) Met-Enkephalin	(3) Met-Enkephalin + 10×excess ICI-174,864	(2)-(1) ΔBRET	n
DOR +	no Gβγ	no Gα	0.0869±0.0004	0.0845±0.0004	0.0823±0.0004	-0.0024	3
GRK-Luc	Venus-Gβγ	Gai3	0.1265±0.0083	0.1735±0.0055	0.1236±0.0072	0.0470	3
		Gaiqi	0.1096±0.0055	0.1540±0.0097	0.1082±0.0061	0.0444	3

			(1) Ligand free	(2) Met-Enkephalin	(3) Met-Enkephalin + 10×excess ICI-174,864	(2)-(1) ΔBRET	n
DOR +	no Gβγ	no Gα	0.0874±0.0008	0.0852±0.0008	0.0827±0.0005	-0.0022	3
GIRK-Luc	Venus-Gβγ	Gai3	0.1006±0.0020	0.1101±0.0022	0.0987±0.0014	0.0095	3
		Gaiqi	0.0945±0.0009	0.1004±0.0022	0.0918±0.0013	0.0059	3

			(1) Ligand free	(2) Dopamine	(3) Dopamine + 10×excess Haloperidol	(2)-(1) ΔBRET	n
D2R +	no Gβγ	no Gα	0.0881±0.0003	0.0863±0.0001	0.0833±0.0003	-0.0018	3
GRK-Luc	Venus-Gβγ	Gai3	0.1304±0.0035	0.1643±0.0057	0.1255±0.0029	0.0339	7
		Gaiqi	0.1215±0.0024	0.1487±0.0041	0.1192±0.0020	0.0272	7
		Gaqi5	0.1115±0.0009	0.1371±0.0032	0.1063±0.0009	0.0256	9
		Gaqiqi5	0.1255±0.0019	0.1638±0.0039	0.1219±0.0018	0.0383	7

			(1) Ligand free	(2) Dopamine	(3) Dopamine + 10×excess Haloperidol	(2)-(1) ΔBRET	n
D2R +	no Gβγ	no Gα	0.0885±0.0001	0.0863±0.0001	0.0834±0.0002	-0.0022	3
GIRK-Luc	Venus-Gβγ	Gai3	0.1046±0.0019	0.1147±0.0020	0.1005±0.0014	0.0101	7
		Gaiqi	0.1023±0.0021	0.1066±0.0021	0.0972±0.0016	0.0043	7
		Gaqi5	0.0968±0.0006	0.0993±0.0016	0.0913±0.0005	0.0025	9
		Gaqiqi5	0.1033±0.0009	0.1122±0.0009	0.1010±0.0007	0.0089	7

* indicates the conditions where the activation of G proteins was not observed.

We also changed the Fig. 1A,C to be consistent with the model shown in Fig. 5a, as follows.

Figure 1 Measuring $G\alpha$ specificity in $G\beta\gamma$ -GIRK binding. (A) Schematic representation of the BRET assay to measure G protein activation. Upon adding receptor agonists, the Venus-tagged $G\beta\gamma$, the BRET acceptor, dissociates from $G\alpha$ and then associates with the BRET donor GRK-Luc, leading to the increased BRET signal. (C) Schematic representation of the BRET assay to measure GIRK- $G\beta\gamma$ binding.

Regarding the basal BRET value for the $G\alpha\beta\gamma$ -GIRK interaction, we cannot simply compare the basal BRET values to evaluate the $G\alpha\beta\gamma$ -GIRK interaction. This is because we expressed the Venus- $G\beta\gamma$ proteins at concentrations slightly higher than those of the $G\alpha$ proteins, to exclude the possibility that the released Venus- $G\beta\gamma$ proteins are recaptured by free $G\alpha$ proteins. Therefore, the experimentally observed basal BRET values reflect not only the interaction between Venus- $G\alpha\beta\gamma$ and GIRK-Luc, but also the interaction between the residual free Venus- $G\beta\gamma$ and GIRK-Luc.

To quantitatively compare the basal BRET between Venus- $G\alpha\beta\gamma$ and GRK/GIRK-Luc before the agonist stimulation, we conducted additional BRET experiments in which an excess amount to $G\alpha$ is expressed, to suppress the basal BRET caused by the interaction between free Venus- $G\beta\gamma$ and GRK-Luc/GIRK-Luc. The basal BRET between Venus- $G\beta\gamma$ and GRK-Luc/GIRK-Luc decreased as the expression level of $G\alpha i3$ increased, and the BRET value nearly reached a plateau at the highest expression level of $G\alpha i3$. These plateau values are expected to reflect the basal BRET between Venus- $G\alpha\beta\gamma$ and GRK-Luc/GIRK-Luc. At the highest $G\alpha$ expression level, the basal BRET values were calculated to be 0.018 for GRK-Luc, and 0.007 for GIRK-Luc, indicating that the basal BRET level between Venus- $G\alpha\beta\gamma$ and GIRK-Luc is not larger than that between Venus- $G\alpha\beta\gamma$ and GRK-Luc.

Supporting Figure 8. BRET experiments observing the basal BRET between G $\alpha\beta\gamma$ and GRK/GIRK. (A) Schematic representation of a competitive binding experiment observing the basal BRET between Venus-G $\alpha\beta\gamma$ and GRK-Luc. (B) Comparisons of the basal BRET values of cells expressing GRK-Luc, Venus-G $\beta\gamma$, and G α i3. When the G α i3 expression level was varied, the amount of the transfected DNA encoding G α i3 was varied between 0 and 570 ng, while the amounts of the transfected DNA encoding GRK-Luc and Venus-G $\beta\gamma$ were fixed to 18 ng and 24 ng, respectively. We calculated the basal BRET between Venus-G $\alpha\beta\gamma$ and GRK-Luc from the difference between the BRET value at the highest G α i3 expression level, and that of GRK-Luc alone. (C) Schematic representation of a competitive binding experiment observing the basal BRET between Venus-G $\alpha\beta\gamma$ and GIRK-Luc. (D) Comparisons of the basal BRET values of cells expressing GIRK-Luc, Venus-G $\beta\gamma$, and G α i3. When the G α i3 expression level was varied, the amount of the transfected DNA encoding G α i3 was varied between 0 and 570 ng, while the amounts of the transfected DNA encoding GIRK1, GIRK2-Luc, and Venus-G $\beta\gamma$ were fixed to 18 ng, 18 ng, and 24 ng, respectively. We calculated the basal BRET between Venus-G $\alpha\beta\gamma$ and GIRK-Luc from the difference between the BRET value at the highest G α i3 expression level, and that of GIRK-Luc alone.

We presume that these results indicate that the basal BRET between Venus-G $\alpha\beta\gamma$ and GIRK-Luc is so small that we could not clearly observe it in our experimental conditions. As the G $\alpha\beta\gamma$ -GIRK interaction is supposed to be about 10-fold weaker than the G $\beta\gamma$ -GIRK interaction, the basal BRET value between Venus-G $\alpha\beta\gamma$ and GIRK-Luc is also expected to be substantially smaller than that observed between Venus-G $\beta\gamma$ and GIRK-Luc (about 0.042). The slightly larger basal BRET between Venus-G $\alpha\beta\gamma$ and GRK-Luc is probably due to the fact that the binding affinity between G $\beta\gamma$ and GRK (30 nM, (Pitcher et al., 1992)) is stronger than that between G $\beta\gamma$ and GIRK (250 μ M, (Yokogawa et al., 2011)), and hence the expressed G α did not completely displace GRK-Luc.

In conclusion, although we expected to observe a higher basal BRET between Venus-Gαβγ and GIRK-Luc in the inactive state, this basal BRET value was too small to quantitatively evaluate in our experimental conditions, due to the inherently low binding affinity between Gαβγ and GIRK.

3-3) Likewise, it is unclear whether the deltaBRET between Gβγ/GIRK monitors the extent of GIRK activation. A small deltaBRET upon agonist application may, e.g. also reflect high tonic GIRK activity (high basal BRET) that cannot be stimulated by agonist to the same extent as in the absence of tonic activity (Naloxone is a neutral antagonist and therefore does not reveal tonic activity in the system). Given these caveats of the BRET assay, I believe that a direct electrophysiological read-out for GIRK activity should be used to, at least, validate the main conclusions of the manuscript (see below).

We appreciate the reviewer's comment. We confirmed that the ΔBRET upon agonist application mainly reflects the extent of GRK or GIRK activation, and is not strongly affected by the high tonic activity, on the basis of the following results.

(i) Regarding the ΔBRET upon the activation of DOR, we measured the BRET values upon the addition of the inverse agonist, ICI-174,864, followed by the addition of the agonist, Met-Enkephalin. Upon the addition of the inverse agonist, the BRET values reversibly returned to the same level as observed before the addition of the agonist. Therefore, the basal BRET value does not reflect the high tonic activity of DOR.

Supporting Figure 9. Plots of the BRET values in the absence of GPCR ligands (black), after the addition of the agonist, Met-Enkephalin (red), and after the addition of an excess amount of the inverse agonist, ICI-174,864 (blue). (Left) Plots of the BRET signal between Gβγ and GRK with different Gα chimeric proteins. (Right) Plots of the BRET signal between Gβγ and GIRK with different Gα chimeric proteins. DOR was co-expressed in all experiments. The BRET values were summarized in Supplementary Table 1.

(ii) Regarding the ΔBRET upon the activation of D2R, we also measured the BRET values upon the addition of the inverse agonist, haloperidol, followed by the addition of the agonist, dopamine. Upon the addition of the inverse agonist, the BRET values reversibly returned to the same level as

observed before the addition of the agonist. These results support the notion that the high basal BRET is not observed, and that the Δ BRET upon the application of the agonist faithfully monitors the extents of GRK or GIRK activation.

Supporting Figure 10. Plots of the BRET values in the absence of GPCR ligands (black), after the addition of the agonist, dopamine (red), and after the addition of an excess amount of the inverse agonist, haloperidol (blue). (Left) Plots of the BRET signal between $G\beta\gamma$ and GRK with different $G\alpha$ chimeric proteins. (Right) Plots of the BRET signal between $G\beta\gamma$ and GIRK with different $G\alpha$ chimeric proteins. D2R was co-expressed in all experiments. The BRET values are summarized in Supplementary Table 1.

We summarized the Δ BRET values in the presence of the inverse agonist in Supplementary Table 1, and added the explanations of the basal activities as follows.

(Page 7, lines 128-136)

The observed increase in the BRET signal (Δ BRET) was reversibly decreased to the basal level by the addition of Naloxone (DOR antagonist), showing that the observed BRET change reflects the binding between GRK and $G\beta\gamma$, controlled by the DOR-mediated G protein signaling pathway. Similar changes in BRET signals were also observed in the cells expressing GIRK-Luc (Figure 1C and D). **These ligand-dependent changes in BRET were good indicators of the extents of GRK or GIRK activation. We confirmed that the effects of the basal activities on the measured BRET values were small, because further decreases in the BRET signal were not observed upon the application of inverse agonists of GPCR (Supplementary Table 1).**

3-4) The most important finding is that the αA helix in the helical domain of Gai forms the binding interface with GIRKs. The only validation that is provided is a loss-of-function experiment showing a decline in the specificity factor for Gai3-q(αA), which was significantly smaller than that of the wild-type Gai3. It would be more convincing to use a gain-of-function experiment to support the main conclusion of the manuscript and to

show that Gaq (or Gas) with the αA helix of Gai shows an increase in the specificity factor.

We appreciate the reviewer's suggestion. According to the reviewer's suggestion, we conducted the BRET experiments of the Gaqi5-i3(αA), Gaqi5 protein with the the αA helix of Gai, to test whether the αA helix of Gai causes an increase in the specificity factor. Unfortunately, we were not able to observe the D2R agonist-dependent $\Delta BRET_{GRK}$ in cells transfected with the Gaqi5-i3(αA) plasmid, indicating that the Gaqi5-i3(αA) proteins were not expressed at a sufficient level and/or Gaqi5-i3(αA) was not coupled to D2R. Therefore, we were not able to evaluate the specificity factor of Gaqi5-i3(αA).

Supporting Figure 11 (A) Topological representations of the Gai3 and Gaqi5-i3(αA) proteins. GD, GTPase domain; HD, helical domain. (B) Plots of the $\Delta BRET$ values between Venus-G $\beta\gamma$ and GRK-Luc ($\Delta BRET_{GRK}$), upon the addition of the D2R agonist, dopamine. (Left) Plot of the $\Delta BRET_{GRK}$ value, using the cells transfected with 285 ng of the Gai3 DNA. (Right) Plots of the $\Delta BRET_{GRK}$ values, using the cells transfected with 95, 190, 380, and 570 ng of the Gaqi5-i3(αA) DNA.

Although it was technically difficult to evaluate the specificity factor of Gaqi5-i3(αA), we designed and assayed another chimeric G α , Gaqiqi5, to further confirm that the helical domain of Gai3 confers the specificity. In Gaqiqi5, the helical domain and the C-terminal region were both replaced with those of Gai3, and the other regions were derived from Gq. Our BRET experiments with Gaqiqi5 showed that the specificity factor of Gaqiqi5 (0.236 ± 0.016) is not significantly different from that of Gai3, and larger than that of Gaqi5 or Gaiqi. These results indicate that the helical domain of Gai3 rescues the

inability to activate GIRK in $G\alpha_{qi5}$, and that $G\alpha_{qiqi5}$ is actually a gain-of-function mutant.

Figure 1 Measuring $G\alpha$ specificity in $G\beta\gamma$ -GIRK binding. (E) Top, the crystal structure of $G\alpha_{i1}$ from $G\alpha_{i1}\beta\gamma$ (PDB ID: 1GP2) (Wall et al., 1995). Bottom, topological representations of the $G\alpha$ proteins used in this study. GD, GTPase domain; HD, helical domain. (F) The $\Delta BRET_{GIRK}/\Delta BRET_{GRK}$ ratios, named specificity factors, for each combination of GPCR and $G\alpha$. Data are means \pm SEM. The **number of measurements taken from independently transfected cell batches is indicated in the bar**. * $p < 0.001$ by unpaired one-sided t test. **Source data are provided as a Source Data File.**

Regarding the gain-of-function mutant, $G\alpha_{qiqi5}$, we added the result of the BRET experiment of $G\alpha_{qiqi5}$ as follows.

(Pages 8-9, lines 163-175)

We also compared the specificity factors using the G_i/o -coupled dopamine D_2 receptor (D2R), and obtained values of 0.303 ± 0.025 , 0.091 ± 0.022 , and 0.157 ± 0.015 for $G\alpha_{i3}$, $G\alpha_{qi5}$, and $G\alpha_{qiqi}$, respectively ($n=7-9$), and the values obtained with $G\alpha_{qi5}$ and $G\alpha_{qiqi}$ were significantly smaller than that obtained with $G\alpha_{i3}$ ($p < 0.001$) (Figure 1F). **To gain further insights into the role of the helical domain, we also prepared a chimeric $G\alpha$, $G\alpha_{qiqi5}$, in which the helical domain of $G\alpha_{qi5}$ (residues 69-180) is replaced with that from $G\alpha_{i3}$ (residues 63-175), and found that the specificity factor of $G\alpha_{qiqi5}$ (0.236 ± 0.016) was significantly larger than that of $G\alpha_{qi5}$, and similar to that of $G\alpha_{i3}$.** These results show that the $G\beta\gamma$ dissociated from $G\alpha_{i3}$ or **$G\alpha_{qiqi5}$** binds to GIRK with significantly higher specificity than the $G\beta\gamma$ dissociated from $G\alpha_{qiqi}$ and $G\alpha_{qi5}$, even though the released $G\beta\gamma$ is identical, and this preference of $G\alpha$ is commonly observed in both the DOR- and D2R-mediated pathways.

3-5) Moreover, gain- and loss-of-function mutants should also be analyzed using NMR and GIRK nanodiscs. This will reveal whether the mutants exhibit the expected shifts in intensities.

We appreciate the reviewer's suggestion. We conducted the NMR experiments to observe the loss-of-function mutant, Gai3-q(α A), and analyzed the NMR spectral changes upon the addition of the GIRK-chimera nanodiscs. As expected, we did not observe significant intensity reductions in Gai3-q(α A) upon the addition of the GIRK-chimera nanodiscs, indicating that Gai3-q(α A) does not interact with GIRK. These results are consistent with the BRET results, in which the specificity factor significantly decreased in Gai3-q(α A), and support our structural model in which the α A helix forms the binding surface for GIRK.

In the revised manuscript, we added the results of the NMR experiment to observe Gai3-q(α A), as follows.

Supplementary Figure 10 NMR spectral changes of Gai3-q(α A)[ILVA]βγ induced by the GIRK chimera-nanodiscs. Plots of relative intensities of Gai3[ILVA]βγ (A) and Gai3-q(α A) [ILVA]βγ (B) upon the addition of 2 eq. of the GIRK chimera-nanodiscs. The error bars are calculated based on the signal-to-noise ratios. Methyl groups with relative intensities lower than 0.80 are colored according to their values. The Gai3[ILVA]βγ data are the same as those shown in Figure 2. Source data are provided as a Source Data File.

(Page 16-17, lines 349-359)

The specificity factors of these chimeras are shown in Figure 4C. The specificity factor for Gai3-q(α A) was 0.159 ± 0.008 ($n=4$), which was significantly smaller than that of the wild-type Gai3 (0.286 ± 0.009 , $p < 0.001$). In contrast, the specificity factors of Gai3-q(α B) and Gai3-q(α E) were 0.248 ± 0.023 ($n=3$, $p=0.041$ vs. Gai3) and 0.249 ± 0.011 ($n=5$, $p=0.022$ vs. Gai3) respectively, which were statistically not significantly

different from that of the wild-type *Gai3*. We also conducted NMR experiments to observe *Gai3-q(αA)[ILVA]βγ*, and significant intensity reductions were not found upon the addition of the GIRK chimera-nanodiscs, indicating that the specific binding to the GIRK chimera was diminished in *Gai3-q(αA)[ILVA]βγ* (Supplementary Figure 10). From these results, we concluded that the αA helix is the key structural element of *Gai/o* that couples specifically with GIRK.

Unfortunately, our attempts to conduct the NMR experiments using the gain-of-function mutant, *Gaiqi5*, have failed, because we were not able to prepare isotopically-labeled proteins of this mutant. This is probably because the GTPase domain of this mutant is derived from *Gq*, which is known to be very difficult to express at high levels in an *E.coli* expression system (Tesmer et al., 2005).

3-6) Given that the BRET experiments only provide an indirect measure for GIRK activation, it is crucial to directly measure whether gain- and loss-of- function mutants activate K⁺ currents in a heterologous expression system. This will give the direct information on whether the mutants activate GIRKs and/or influence current activation kinetics in the expected manner.

We appreciate the reviewer's suggestion. The electrophysiological recording of the GIRK activity may be an alternative approach to make the results more convincing. However, we believe that our BRET results provide the necessary and sufficient evidence to validate our conclusions, for the following reasons.

(i) Our BRET results are highly consistent with those obtained from the previously reported electrophysiological experiments. Rusinova and co-workers previously conducted electrophysiological analyses of GIRK using *Gai3*, *Gaiqi*, and *Gaiqi5*, and showed similar trends to those we observed in our BRET experiments (Rusinova et al., 2007). These results strongly support the robustness of our approach, and hence it is highly expected that the newly designed *Ga* mutants in our study would show consistent electrophysiological properties. The consistency of the electrophysiological recordings of GIRK and the BRET experiments has been also reported in recently published results (Masuho et al., 2015; Touhara and MacKinnon, 2018).

[redacted]

Supporting Figure 12 Comparison of the results of electrophysiological analyses of the GIRK current (left) and the results of our BRET experiments observing the binding of *Gβγ* to GIRK (right).

The results of electrophysiological recordings of GIRK are modified from the figures in the paper (Rusinova et al., 2007). Plots of the normalized currents of GIRK4 upon the addition of the M2R agonist, acetylcholine, are shown.

(ii) In the BRET experiments, we can quantitatively evaluate the intrinsic activities and the binding properties of $G\alpha$ proteins, which are necessary to interpret the results from the NMR experiments obtained with the purified proteins. In the electrophysiological analyses, the decrease in the GIRK current can be induced not only by the decrease in the extent of GIRK activation, but also by the impaired coupling to GPCR and/or impaired activation of $G\alpha$ itself. In our BRET experiments, the latter two possibilities can be examined from the positive control experiments observing the BRET between $G\beta\gamma$ and GRK. (Actually, we were able to successfully identify a dysfunctional mutant of $G\alpha$, $G\alpha_q(\alpha A-i3)$, from these positive control experiments.) Therefore, although we admit that the electrophysiological experiments would provide valuable information regarding the extent of the GIRK activation, the BRET experiments have their own advantages over the electrophysiological analyses, and are more suitable to combine with the results of the NMR experiments.

3-7) I am surprised that no larger spectral changes were observed with $Gaiqi[ILVA]\beta\gamma$ in the N-terminal region, which I would expect to interact with membrane lipids of the nanodiscs even though $Gaiqi[ILVA]\beta\gamma$ does not bind GIRKs (Fig. 2C).

We appreciate the reviewer's comment. This is because the population of $Gaiqi[ILVA]\beta\gamma$ bound to the nanodisc membrane is so small that we could not detect the NMR spectral changes. In order to stably form the $G\alpha\beta\gamma$ -GIRK complex, $G\alpha\beta\gamma$ must interact with both GIRK and the nanodisc membrane. In the case of the $Gaiqi[ILVA]\beta\gamma$ -GIRK-nanodisc interaction, GIRK does not contribute to the complex formation, and hence the bound population of $Gaiqi[ILVA]\beta\gamma$ is expected to be very small. This notion is also supported from the result that we did not observe significant NMR spectral changes of $Gai3[ILVA]\beta\gamma$ upon the addition of empty nanodiscs, reflecting the fact that the interaction with GIRK significantly contributes to the complex formation.

(Please also refer to our responses to 1-4 and 2-5-2)

Minor

3-8) The abstract needs rewriting, as it is too generic and does not clearly highlight the novel insights this study provides. Several labs have shown that Gao/i binds GIRK subunits and that this interaction contributes to the specificity of GIRK activation. Likewise, the $G\alpha$ helical domain was implicated in this interaction before. This is not really novel and should therefore not be the main conclusion of the abstract. In my opinion, the most significant finding is the identification of a binding interface, which should be clearly stated in the abstract (and accordingly thoroughly validated, see above).

We appreciate the reviewer's suggestion. We rewrote the abstract to clarify the most important findings in our study: the identification of the binding surface and the construction of a structural model of the $G_{\alpha i}/\beta\gamma$ -GIRK complex.

(Abstract)

G protein-gated inwardly rectifying potassium channel (GIRK) plays a key role in regulating neurotransmission. GIRK is opened by the direct binding of the G protein $\beta\gamma$ subunit ($G\beta\gamma$), which is released from the heterotrimeric G protein ($G\alpha\beta\gamma$) upon the activation of G protein-coupled receptors (GPCRs). GIRK contributes to precise cellular responses by specifically and efficiently responding to the G_i/o -coupled GPCRs. However, the detailed mechanisms underlying this family-specific and efficient activation are largely unknown. Here, we investigated the structural mechanism underlying the G_i/o family-specific activation of GIRK, by combining cell-based BRET experiments and NMR analyses in a reconstituted membrane environment. We revealed that the interaction formed by the αA helix of $G_{\alpha i}/o$ mediates the formation of the $G_{\alpha i}/\beta\gamma$ -GIRK complex, which is responsible for the family-specific activation of GIRK. We also constructed a model structure of the $G_{\alpha i}/\beta\gamma$ -GIRK complex, which provides the molecular basis underlying the specific and efficient regulation of GIRK. (150 words)

3-9) P3 “...thus regulating heart rate and inhibitory neurotransmission”. GIRKs regulate excitatory neurotransmission as well, e.g. by shunting of EPSCs

We appreciate the reviewer's comment. We rewrote the sentence as follows.

(Page 3, lines 42-44)

Therefore, the outward K^+ current induced by the opening of GIRK hyperpolarizes the membrane and decreases cell excitabilities, thus regulating the heart rate and **both the excitatory and** inhibitory neurotransmissions.

Reference

- Denisov, I.G., Grinkova, Y. V., Lazarides, A.A., and Sligar, S.G. (2004). Directed Self-Assembly of Monodisperse Phospholipid Bilayer Nanodiscs with Controlled Size. *J. Am. Chem. Soc.* 126, 3477–3487.
- Frey, L., Lakomek, N.A., Riek, R., and Bibow, S. (2017). Micelles, Bicelles, and Nanodiscs: Comparing the Impact of Membrane Mimetics on Membrane Protein Backbone Dynamics. *Angew. Chemie - Int. Ed.* 56, 380–383.
- Grasberger, B., Minton, A.P., DeLisi, C., and Metzger, H. (1986). Interaction between proteins localized in membranes. *Proc. Natl. Acad. Sci. U. S. A.* 83, 6258–6262.

Hollins, B., Kuravi, S., Digby, G.J., and Lambert, N.A. (2009). The c-terminus of GRK3 indicates rapid dissociation of G protein heterotrimers. *Cell. Signal.* 21, 1015–1021.

Mase, Y., Yokogawa, M., Osawa, M., and Shimada, I. (2012). Structural basis for modulation of gating property of G protein-gated inwardly rectifying potassium ion channel (GIRK) by *i/o*-family G protein α subunit ($G\alpha_{i/o}$). *J. Biol. Chem.* 287, 19537–19549.

Masuho, I., Ostrovskaya, O., Kramer, G.M., Jones, C.D., Xie, K., and Martemyanov, K.A. (2015). Distinct profiles of functional discrimination among G proteins determine the actions of G protein-coupled receptors. *Sci. Signal.* 8, ra123-ra123.

Maziarz, M., and Garcia-Marcos, M. (2017). Rapid kinetic BRET measurements to monitor G protein activation by GPCR and non-GPCR proteins (Elsevier Inc.).

Pitcher, J.A., Inglese, J., Higgins, J.B., Arriza, J.L., Casey, P.J., Kim, C., Benovic, J.L., Kwatra, M.M., Caron, M.G., and Lefkowitz, R.J. (1992). Role of beta gamma subunits of G proteins in targeting the beta-adrenergic receptor kinase to membrane-bound receptors. *Science* 257, 1264–1267.

Rusinova, R., Mirshahi, T., and Logothetis, D.E. (2007). Specificity of $G\beta\gamma$ signaling to Kir3 channels depends on the helical domain of pertussis toxin-sensitive $G\alpha$ subunits. *J. Biol. Chem.* 282, 34019–34030.

Sarvazyan, N.A., Remmers, A.E., and Neubig, R.R. (1998). Determinants of $G_{i1}\alpha$ and $\beta\gamma$ binding. Measuring high affinity interactions in a lipid environment using flow cytometry. *J. Biol. Chem.* 273, 7934–7940.

Tesmer, V.M., Kawano, T., Shankaranarayanan, A., Kozasa, T., and Tesmer, J.J.G. (2005). Snapshot of activated G proteins at the membrane: the $G\alpha_q$ -GRK2-Gbetagamma complex. *Science* (80-.). 310, 1686–1690.

Touhara, K.K., and MacKinnon, R. (2018). Molecular basis of signaling specificity between GIRK channels and GPCRs. *Elife* 7, 399–404.

Wall, M.A., Coleman, D.E., Lee, E., Iñiguez-Lluhi, J.A., Posner, B.A., Gilman, A.G., and Sprang, S.R. (1995). The structure of the G protein heterotrimer $G_{i\alpha1}\beta_1\gamma_2$. *Cell* 83, 1047–1058.

Whorton, M.R., and MacKinnon, R. (2013). X-ray structure of the mammalian GIRK2- $\beta\gamma$ G-protein complex. *Nature* 498, 190–197.

Yokogawa, M., Osawa, M., Takeuchi, K., Mase, Y., and Shimada, I. (2011). NMR analyses of the $G\beta\gamma$ binding and conformational rearrangements of the cytoplasmic pore of G protein-activated inwardly rectifying potassium channel 1 (GIRK1). *J. Biol. Chem.* 286, 2215–2223.

Yokogawa, M., Kobashigawa, Y., Yoshida, N., Ogura, K., Harada, K., and Inagaki, F. (2012). NMR Analyses of the Interaction between the FYVE Domain of Early Endosome Antigen 1 (EEA1) and Phosphoinositide Embedded in a Lipid Bilayer. *J. Biol. Chem.* 287, 34936–34945.

REVIEWERS' COMMENTS:

Reviewer #1 (Remarks to the Author):

In the revised version of the paper, the authors addressed all my comments and clarified my open questions. I appreciate the detailed response and the amount of changes made as well as the additional data included in the manuscript. Thus, I recommend publication of the paper in its current form.

Reviewer #2 (Remarks to the Author):

The revised manuscript has addressed all the outstanding questions quite comprehensively. The authors have conducted further experiments and have extended discussions accordingly in the main text as well as in the supplementary information. I am happy with their revisions and consider the manuscript as publishable as it stands.

Reviewer #3 (Remarks to the Author):

The authors have adequately addressed most of the points that I raised. However, there are few remaining concerns that need to be addressed prior to publication.

The read-out of the Gbg-GIRK BRET may differ from the recording of Kir3 currents because the BRET assay is influenced by overexpression of G-protein subunits. For example, the paper by Touhara and MacKinnon shows that overexpression of Gai increases Gbg-GIRK BRET because exogenous G protein subunits allow more Gbg to be generated. In contrast overexpression of Gai decreases Kir3 currents because endogenous Gbg is scavenged. BRET and K-currents therefore provide complementary information and should be used in parallel. The main constructs (Gai3, Gaqi5, Gai3-q(aA), Gaqiqi5) should therefore be verified by measuring Kir3 currents. The BRET data show that these Ga constructs are neither impaired in their coupling to GPCR nor impaired in activation.

When performing multiple comparison the authors must use ANOVA instead of t-tests (e.g. Fig. 1F and 4C). This may influence conclusions.

This paper is in clear contrast to the paper by Touhara and MacKinnon that explains the specificity of signaling with an intrinsically higher rate of G protein association with the GPCR rather than by a specific binding-site for Gai on GIRK channels. In the model by Touhara and MacKinnon hot-spots of higher GPCR and G protein density are only required to achieve GIRK-activating concentrations of Gbg but not to provide the specificity of signaling. The two proposed mechanisms are therefore fundamentally different, which should be clearly stated in the Discussion section.

Point-by-point responses to the reviewer

We would like to thank the reviewer for the positive comments and suggestions on our manuscript. According to the reviewer's comments, we have revised the manuscript carefully. Our point-by-point responses are summarized below.

Reviewer 3

The authors have adequately addressed most of the points that I raised. However, there are few remaining concerns that need to be addressed prior to publication.

1) The read-out of the Gbg-GIRK BRET may differ from the recording of Kir3 currents because the BRET assay is influenced by overexpression of G-protein subunits. For example, the paper by Touhara and MacKinnon shows that overexpression of Gai increases Gbg-GIRK BRET because exogenous G protein subunits allow more Gbg to be generated. In contrast overexpression of Gai decreases Kir3 currents because endogenous Gbg is scavenged. BRET and K-currents therefore provide complementary information and should be used in parallel. The main constructs (Gai3, Gai3-q(α A), Gai3-q(α A), Gai3-q(α A), Gai3-q(α A)) should therefore be verified by measuring Kir3 currents. The BRET data show that these Ga constructs are neither impaired in their coupling to GPCR nor impaired in activation.

We appreciate the reviewer's thoughtful comments.

The reviewer pointed out that the effect of the overexpression of $G\alpha$ can be different in the electrophysiological experiment and in the BRET assay, referencing the results recently reported by Touhara and MacKinnon (eLife 2018;7:e42908. DOI: <https://doi.org/10.7554/eLife.42908>). However, the observed differences are actually attributed to the differences in the experimental conditions, and the results from electrophysiological experiments and BRET assays are highly consistent even under the conditions with the overexpression of $G\alpha$.

We note that the effect of the overexpression of $G\alpha$ is strongly dependent on whether $G\beta\gamma$ is also overexpressed or not. The observed effects on electrophysiological recordings and Δ BRET values are summarized in Supporting Table below, which includes the results of our additional experiments and the results reported by Touhara and MacKinnon. When the expression level of $G\alpha$ alone is increased, both the agonist-induced GIRK current and the Δ BRET between GIRK and $G\beta\gamma$ decreased, reflecting that $G\beta\gamma$ is sequestered by free $G\alpha$. On the other hand, when the expression levels of both $G\alpha$ and $G\beta\gamma$ are increased, both the GIRK current and the Δ BRET increased, reflecting the increased amount of the available $G\alpha\beta\gamma$. These results strongly support our notion that the electrophysiological experiment and BRET assay are highly consistent with each other, and Touhara and MacKinnon actually pointed out the consistency in the paper (Page 6, "The electrophysiological and BRET assays are in complete agreement with each other").

Editorial Note: Figures in the Supporting Table below are modified from Touhara, K.K., and MacKinnon, R. (2018). Molecular basis of signaling specificity between GIRK channels and GPCRs. *Elife* 7, p.399–404. eLife 2018;7:e42908 DOI: 10.7554/eLife.42908. License: <https://creativecommons.org/licenses/by/4.0/>

Supporting Table. The effects of overexpression of Gα and Gβγ

	Gα overexpression	Gα + Gβγ overexpression
K ⁺ current	A M2R/GIRK stable HEK cells ACh-activated current (pA/pF) Ctrl (N = 4) + Gα₁ (N = 5) Modified from Figure 3A of Touhara & MacKinnon	A M2R/GIRK stable HEK cells ACh-activated current (pA/pF) Ctrl (N = 4) + Gα₁ + Gβγ (N = 4) Modified from Figure 3A of Touhara & MacKinnon
BRET	ΔBRET_{GIRK} Gai3 DNA (ng) Supporting Figure Gai3 DNA amount dependency of ΔBRET_{GIRK}. The DNA amounts of Venus-Gβγ and GIRK1/2-Luc were fixed to 24 ng and 18 ng, respectively. PTX was co-expressed to inhibit activation of endogenous Gai/o.	E M2R + Gβγ-Venus + GIRK-NLuc Normalized ΔBRET ratio 0×Gα₁ 1×Gα₁ 2×Gα₁ 4×Gα₁ (N = 6) Modified from Figure 3-supplement 1 of Touhara & MacKinnon

The reviewer may concern that the overexpression of both Gα and Gβγ in our BRET assays do not faithfully reflect physiological conditions. To investigate this possibility, we conducted the BRET experiments with different expression levels of Gαβγ (Supplementary figure 3). We found that the difference in specificity factor between Gai3 and Gaiq1 became more prominent as the total expression level of Gαβγ decreased. These results suggest that the difference in specificity factor would be more prominent under the physiological condition, where the expression level of Gαβγ is far lower than that in the BRET assays, strongly supporting our notion that the helical domain of Gα is a major determinant defining the family specificity in activating GIRK.

Supplementary Figure 3 Effects of the amount of expressed $G\alpha\beta\gamma$ on the selectivity factors. The selectivity factors ($\Delta\text{BRET}_{\text{GIRK}}/\Delta\text{BRET}_{\text{GRK}}$ ratios) were measured in cells expressing different amounts of $G\alpha i3\beta\gamma$ or $G\alpha iqi\beta\gamma$ ($n=4-10$). For each of the indicated DNA amounts of Venus- $G\beta\gamma$, the DNA amounts of $G\alpha$, GRK-Luc, GIRK1, and GIRK2-Luc were optimized and used (see Table 3). The ratios of the average values of i3 to those of iqi are shown in red. Data are means \pm SEM. The number of measurements taken from independently transfected cell batches is indicated in the bar. Source data are provided as a Source Data File.

2) When performing multiple comparison the authors must use ANOVA instead of t-tests (e.g. Fig. 1F and 4C). This may influence conclusions.

We appreciate the reviewer's suggestion. We performed one-way ANOVA with Tukey-Kramer's pairwise multiple comparison as a post hoc test with an overall significance level of 0.001. We still found significant differences between i3-iqi, i3-qi5, qi5-qi5, and i3-i3-q(α A), so the conclusions were not affected. We rewrote the figure legends accordingly.

3) This paper is in clear contrast to the paper by Touhara and MacKinnon that explains the specificity of signaling with an intrinsically higher rate of G protein association with the GPCR rather than by a specific binding-site for Gai on GIRK channels. In the model by Touhara and MacKinnon hot-spots of higher GPCR and G protein density are only required to achieve GIRK-activating concentrations of Gbg but not to provide the specificity of signaling. The two proposed mechanisms are therefore fundamentally different, which should be clearly stated in the Discussion section.

We appreciate the reviewer's suggestion. We rewrote the discussion section to explain the differences between our model and that proposed by Touhara and MacKinnon more clearly.

Several lines of evidence have indicated that the signaling complex also contains GPCR, along with G protein and GIRK¹²⁻¹⁴. The GPCR-binding site on $G\alpha\beta\gamma$ is the $\alpha 5$ helix in the GTPase domain of $G\alpha$ ¹⁵, and since this region is not involved in the interaction with GIRK, GPCR can also bind to $Gai/o\beta\gamma$ in the proposed complex structure (Supplementary Figure 12). These observations suggest that the $Gai/o\beta\gamma$ -GIRK complex revealed here represents part of an even larger complex including GPCR, and the formation of this large signaling complex underlies the $G\alpha$ family specific activation of GIRK. This model is in contrast to the model recently proposed by Touhara and MacKinnon¹⁶, in which the specificity of GIRK signaling is attributed to the difference in the association rate of G protein with GPCR, rather than by a specific binding of $G\alpha\beta\gamma$ to GIRK. Although the origin of the family specificity is different between these two models, we note that these two models commonly assume that the local concentrations of GPCR, GIRK, and G proteins are maintained at high levels to achieve sufficient concentration of $G\beta\gamma$ to invoke the activation of GIRK upon the activation of GPCR. We also note that the mechanisms underlying the $G\alpha$ family specificity may be different between Gai/o versus Gas investigated by Touhara and MacKinnon, and Gai/o versus Gaq investigated in our study.